Citation: *Molecular Systems Biology* 9:652
www.molecularsystemsbiology.com

# SH3 interactome conserves general function over specific form

Xiaofeng Xin[1,2,13,14], David Gfeller[1,13,15], Jackie Cheng[3,13,16], Raffi Tonikian[1,2,13,17], Lin Sun[4,18], Ailan Guo[5], Lianet Lopez[1], Alevtina Pavlenco[1], Adenrele Akintobi[4], Yingnan Zhang[6], Jean-François Rual[7,8,19], Bridget Currell[9], Somasekar Seshagiri[9], Tong Hao[7,8], Xinping Yang[7,8], Yun A Shen[7,8], Kourosh Salehi-Ashtiani[7,8,20], Jingjing Li[1,2], Aaron T Cheng[3], Dryden Bouamalay[3], Adrien Lugari[10], David E Hill[7,8], Mark L Grimes[11], David G Drubin[3], Barth D Grant[4], Marc Vidal[7,8], Charles Boone[1,2,*], Sachdev S Sidhu[1,2,*] and Gary D Bader[1,2,12,*]

[1] The Donnelly Centre, University of Toronto, Toronto, Ontario, Canada, [2] Department of Molecular Genetics, University of Toronto, Toronto, Ontario, Canada, [3] Department of Molecular and Cell Biology, University of California Berkeley, Berkeley, CA, USA, [4] Department of Molecular Biology and Biochemistry, Rutgers University, Piscataway, NJ, USA, [5] Cell Signaling Technology, Danvers, MA, USA, [6] Department of Early Discovery Biochemistry, Genentech, South San Francisco, CA, USA, [7] Center for Cancer Systems Biology (CCSB) and Department of Cancer Biology, Dana-Farber Cancer Institute, Boston, MA, USA, [8] Department of Genetics, Harvard Medical School, Boston, MA, USA, [9] Department of Molecular Biology, Genentech, South San Francisco, CA, USA, [10] IMR Laboratory, UPR 3243, Institut de Microbiologie de la Méditérannée, CNRS and Aix-Marseille Université, Marseille Cedex 20, France, [11] Division of Biological Sciences, Center for Structural and Functional Neuroscience, The University of Montana, Missoula, MT, USA and [12] Department of Computer Science, University of Toronto, Toronto, Ontario, Canada
[13] These authors contributed equally to this work.
[14] Present address: Department of Biological Engineering, Massachusetts Institute of Technology, Cambridge, MA 02139, USA.
[15] Present address: Swiss Institute of Bioinformatics, Molecular Modelling, Génopode, 1015 Lausanne, Switzerland.
[16] Present address: MedImmune, 24500 Clawiter Road, Hayward, CA 94541, USA.
[17] Present address: Department of Translational Sciences, Biogen Idec, Cambridge, MA 02142, USA.
[18] Present address: Department of Physiology and Biophysics, Boston University School of Medicine, Boston, MA 02118, USA.
[19] Present address: Department of Pathology, University of Michigan, Ann Arbor, MI, USA.
[20] Present address: Division of Science and Math, Center for Genomics and Systems Biology, New York University Abu Dhabi, PO Box 129188, Abu Dhabi, UAE.
* Corresponding authors. C Boone or SS Sidhu or GD Bader, The Donnelly Centre, University of Toronto, 160 College Street, #602, Toronto, Ontario, Canada M5S 3E1.
E-mail: charlie.boone@utoronto.ca or E-mail: sachdev.sidhu@utoronto.ca or Tel.: +416 978 3935;
E-mail: gary.bader@utoronto.ca

Src homology 3 (SH3) domains bind peptides to mediate protein–protein interactions that assemble and regulate dynamic biological processes. We surveyed the repertoire of SH3 binding specificity using peptide phage display in a metazoan, the worm *Caenorhabditis elegans,* and discovered that it structurally mirrors that of the budding yeast *Saccharomyces cerevisiae.* We then mapped the worm SH3 interactome using stringent yeast two-hybrid and compared it with the equivalent map for yeast. We found that the worm SH3 interactome resembles the analogous yeast network because it is significantly enriched for proteins with roles in endocytosis. Nevertheless, orthologous SH3 domain-mediated interactions are highly rewired. Our results suggest a model of network evolution where general function of the SH3 domain network is conserved over its specific form.
*Molecular Systems Biology* **9**: 652; published online 2 April 2013; doi:10.1038/msb.2013.9
*Subject Categories:* proteins; signal transduction; membranes & transport
*Keywords:* network evolution; phage display; protein interaction conservation; SH3 domains; yeast two-hybrid

## Introduction

The physical wiring diagram of the cell is rich in protein–protein interactions (PPIs). A specialized subset of PPIs are mediated by peptide recognition modules (PRMs), which bind short linear peptides (Pawson and Nash, 2003). For example, SH3 (Src homology 3) domains typically bind to proline-rich peptides within target proteins (Pawson and Schlessinger, 1993; Ren *et al*, 1993; Tong *et al*, 2002; Zarrinpar *et al*, 2003a; Tonikian *et al*, 2009), and PDZ (PSD-95/Discs-large/ZO-1) domains often bind to hydrophobic C-termini (Ranganathan and Ross, 1997; Stiffler *et al*, 2007; Tonikian *et al*, 2008). PRMs

function to assemble dynamic signaling complexes and are major players in eukaryotic signaling pathways (Pawson and Nash, 2003; Zarrinpar *et al*, 2003a).

SH3 domains are among the most abundant PRMs encoded by eukaryotic genomes. The budding yeast *Saccharomyces cerevisiae* has 27 SH3 domains and this number increases in multicellular organisms, with 84 in the metazoan worm *Caenorhabditis elegans* and close to 300 in human (Finn *et al*, 2008; Letunic *et al*, 2009). While many SH3 domains broadly target two main binding motifs (Class I: +XXPXXP and Class II: PXXPX+, where P is proline, +is arginine or lysine, and X is any amino acid; Zarrinpar *et al*, 2003a), they

have differences in binding preference at other positions surrounding the core PXXP motif important for determining their interaction specificity. Further, more than one third of yeast SH3 domains do not match these two established classes. Thus, a major challenge is to map SH3 domain specificity in detail and construct PPI networks in which every domain is linked to its partners. As shown in yeast, this can be achieved by combining domain specificity profiles and large-scale protein interaction data (Tong *et al*, 2002; Tonikian *et al*, 2009).

To elucidate the general role of SH3 domains in more complex organisms, we mapped the SH3 interactome in *C. elegans*. We first used phage display technology to determine the specificity of worm SH3 domains (Tonikian *et al*, 2007) and found that their binding specificity repertoire appears to be conserved between yeast and worm. We then conducted large-scale stringent yeast two-hybrid (Y2H) screens to identify binding proteins for most worm SH3 domains. Comparing this network with the equivalent map in yeast, we observed functional enrichment in endocytosis proteins in both networks. Surprisingly, however, we observe very low conservation of PPIs between these two maps. This rewiring has several consequences for our understanding of protein interaction network evolution and protein function conservation. Information transfer of function from one protein to another across organisms, traditionally based on sequence or structural similarity, is a central concept in biology. As the function of a protein is largely defined by its molecular interactions, it is important to expand this concept and better understand the interplay between protein interaction conservation and protein function conservation (Ihmels *et al*, 2005; Beltrao and Serrano, 2007; Wang *et al*, 2010; Habib *et al*, 2012; Kim *et al*, 2012). In particular, studying network evolution mechanisms can help us better understand how protein function evolves and more accurately transfer it across organisms. In this work, our data suggest a model where PPIs are rewired to adapt the use of biological processes like endocytosis in organism-specific pathways, such as sporulation in yeast and phagocytosis in worm. Our network also enables us to predict new endocytosis proteins in worm and human. In particular, our analysis reveals a new interaction between AMPH-1 and TBC-2 important for the recruitment of TBC-2 to endosomal membranes and predicts many novel protein participants in human endocytosis and related processes, which we experimentally validate.

# Results

## Mapping the binding specificity of worm SH3 domains

We performed peptide phage display (Tonikian *et al*, 2007) to determine the breadth of the worm SH3 domain binding specificity repertoire and compared it with the one in yeast. We purified 60 of an attempted 84 SH3 domains predicted within the worm genome (Supplementary Table 1), and we used these in binding selections with peptide phage libraries (see Materials and methods). We obtained results for 36 SH3 domains, a success rate similar to that achieved in previous large-scale domain studies (Huang *et al*, 2008; Tonikian *et al*,

2008; Tonikian *et al*, 2009). The peptides binding to each of these domains were then manually aligned, and binding specificities were modeled with position weight matrices (PWMs) (see Materials and methods). To compare the worm data with that previously published for 24 yeast SH3 domains (Tonikian *et al*, 2009), we hierarchically clustered the combined data set of binding specificities (Figure 1, see Materials and methods). Clustering revealed that half of the worm domains (18 of 36) have either class I or class II specificities and are often grouped with yeast domains, while the other half are dual specific (1) or bind atypical motifs (17) and are similar to atypical yeast domains. There are no large clusters consisting exclusively of either worm or yeast SH3 domains. Thus, the SH3 domain binding specificity repertoire appears to be conserved between yeast and worm.

To explore the conservation of binding specificity at the single domain level, we examined the SH3 binding specificity for all paralog and ortholog pairs of yeast and worm SH3 domains with available phage display data (Figure 2). While the four paralog pairs in yeast display a strong conservation at the level of the SH3 domain sequence and binding specificity, this is not the case for the worm paralog pair (TOCA-1—TOCA-2). Similarly, two pairs of orthologous SH3-containing proteins display different SH3 binding specificities (HUM-1—Myo3p/Myo5p and STAM-1—Hse1p), while (K08E3.4—Abp1p) and (SDPN-1—Bzz1p) have roughly conserved specificity (see Figure 2). Domain sequence similarity explains this pattern, as sequence identity is higher in the conserved yeast paralogs ($>60\%$) and lower in the worm paralogs and yeast-to-worm orthologs ($<50\%$), within the background level of similarities of the SH3 domain family (see Supplementary Figure 1). While additional phage display data could theoretically reveal other SH3 orthologs with a conserved specificity, there are no other pairs of worm-to-yeast ortholog SH3 domains with sequence identity $>50\%$, suggesting that their specificity is not necessarily conserved (see Supplementary Figure 1). Moreover, for 6 proteins out of 16 worm or yeast SH3-containing proteins with orthologs in the other species, some of their orthologs do not contain SH3 domains, and hence their binding properties and specificity are clearly not conserved. From this, we can conclude that although completely unique binding specificity profiles are not observed in worm and yeast, the specificity of individual SH3 domains is only partly conserved.

## Mapping the worm SH3 interactome

To better understand the function of worm SH3 domains and investigate the level of conservation of SH3-mediated protein interactions, we mapped their interactions with worm proteins using large-scale stringent Y2H screens (Dreze *et al*, 2010). We first screened 80 worm SH3 domains with both activation domain (AD)-ORFeome (Lamesch *et al*, 2004) and AD-cDNA libraries (Supplementary Table 1). To expand this network, we performed a second round of screening on highly connected proteins identified in the initial screens (Supplementary Table 2). We combined and filtered all resulting PPIs keeping only those supported by two independent lines of evidence: two colonies in the Y2H screen (984 in total), or one yeast colony but known in the literature (19 in total) or containing a sequence motif matching the phage-derived SH3 domain

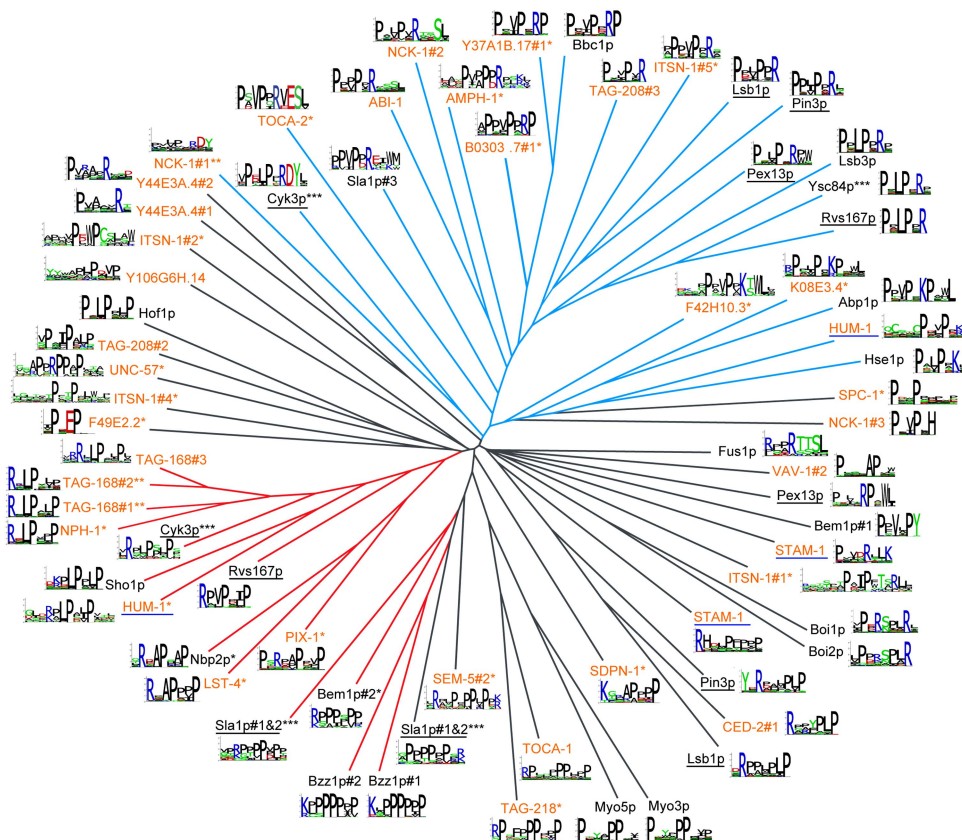

**Figure 1**  The worm and yeast SH3 domain peptide-binding specificity repertoire is conserved. SH3 domain specificities (36 worm, orange labels; 24 yeast, black labels), visualized as sequence logos, are grouped by similarity in a tree (see Experimental Procedures). Tree branches are colored according to their specificity class: I (red), II (blue), and atypical (black). Underlined labels indicate domains that exhibit multiple specificities. Multiple peptide libraries were used to determine specificity: $X_6$-PXXP-$X_6$ (*), $X_6$-PXXP-$X_6$ and $X_{12}$ (**), $X_6$-PXXP-$X_6$ and $X_7$-R/K-$X_7$ (***), or $X_{12}$ (no asterisk), where P is proline, R is arginine, K is lysine, and X is any amino acid (see Experimental Procedures).

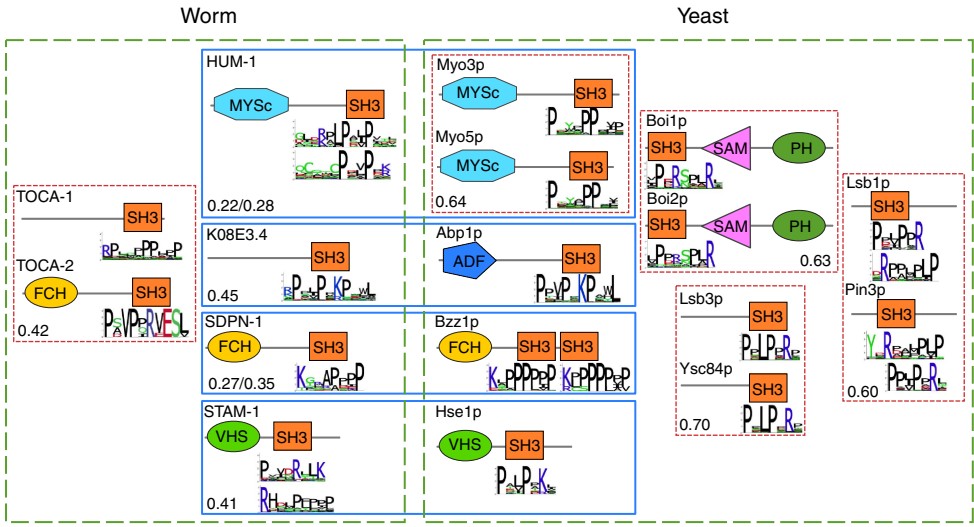

**Figure 2**  SH3 domain binding specificities are conserved between yeast paralogs, but not necessarily between yeast and worm orthologs. Protein domain architecture and SH3 domain binding specificity are shown for the yeast and worm SH3 protein paralogs and orthologs with available peptide phage display data. Yeast and worm SH3 proteins are grouped by green boxes, orthologs are grouped by blue boxes, and paralogs are grouped by red boxes. Lower left corner of boxes shows the sequence identity between the SH3 domains in the box. The domain architecture is defined by SMART (Letunic *et al*, 2009; Schultz *et al*, 2000), but is not drawn to scale. ADF, Actin depolymerization factor/cofilin-like domain; FCH, Fes/CIP4 homology domain; MYSc, myosin ATPases; PH, Pleckstrin homology domain; SAM, Sterile alpha motif; VHS, domain present in VPS-27, Hrs and STAM.

binding specificity (67 in total) (see Materials and methods for more details). Interactions involving transcription factors (TFs) with more than five connections in our network were removed, since these likely correspond to auto-activating preys (see Materials and methods). The resulting network contains 1070 PPIs connecting 79 worm SH3 domains (found in 63 SH3-containing proteins) and 475 proteins (Supplementary Table 3). This network has significant overlap with known PPIs ($P < 2.6 \times 10^{-47}$) (Simonis *et al*, 2009), interologs ($P = 1.2 \times 10^{-37}$) (Walhout *et al*, 2000; Matthews *et al*, 2001; Simonis *et al*, 2009), and functional interactions ($P < 1.2 \times 10^{-28}$) (Lee *et al*, 2008) (both comparisons exclude overlap with interactions included in our network based on the literature evidence). Similarly, the network has higher than expected Gene Ontology (GO) semantic similarity among interacting proteins (Lin, 1998; Supplementary Tables 4 and 5), indicating that it significantly agrees with previous data (see Materials and methods and Supplementary information). Twenty five of these SH3 proteins containing thirty-three SH3 domains have never been reported to interact with any protein (Supplementary Table 6), thus a significant portion of our worm SH3 interactome data set is novel.

Y2H and phage display experiments query different but overlapping regions of PPI space (Tong *et al*, 2002; Tonikian *et al*, 2009). To quantify this overlap, we scanned the worm proteome for proteins containing motifs that match the 36 worm SH3 domain binding specificities determined by phage display, using an established PWM-based scoring algorithm (see Materials and methods) (Tong *et al*, 2002; Tonikian *et al*, 2009). The resulting predicted protein interactions significantly overlap the Y2H interactome ($P < 10^{-6}$), with 18% of SH3-mediated Y2H PPIs among the top 100 predictions and 37% among the top 500 (see Materials and methods and Figure 3, green curve). Plotting the same data for each domain separately (see Supplementary Figure 2), we observe a clearly significant enrichment ($P < 0.001$) for the majority (26 out of

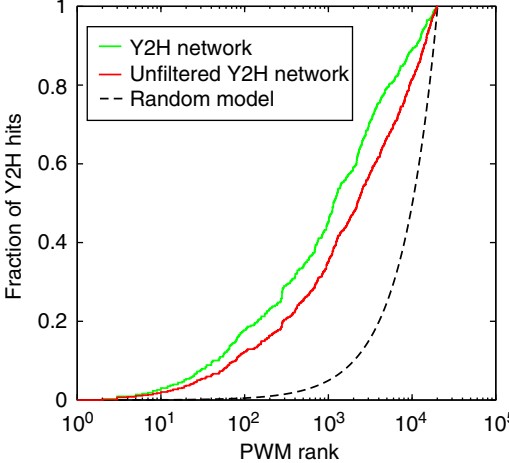

**Figure 3** Overlap between Y2H and phage display predicted PPIs. Each worm SH3 domain with a phage-derived specificity profile represented as a PWM was used to score and rank all worm proteins for matches to this PWM. The plot shows the fraction of PPIs with a rank higher than the value on the *x* axis. Full Y2H is red, filtered Y2H (Supplementary Table 3) is green, and the expected results for randomly distributed PPIs is black. The observed enrichments are all highly significant ($P < 10^{-6}$).

36) of SH3 domains with phage profiles. Domains with lower enrichment between the phage predictions and the Y2H hits typically have fewer interactions (average 4.8 versus 15.3 for the other domains) which may result from experimental limitations, such as incomplete binding information captured by peptide phage display (e.g., larger binding interfaces and domain–domain interactions). Interactions ($n = 67$) part of the Y2H network based on SH3 binding motif presence were not included in phage overlap analysis. Overall, we conclude that the worm SH3 domain interactome mapped by Y2H and the phage-derived binding specificities support each other.

## SH3 domains have a conserved general role in endocytosis

To examine the function of worm SH3 domains and their interactors, we searched for significantly enriched gene function annotation terms in our interactome (see Materials and methods). Functions related to endocytosis, cytoskeleton, and small GTPase signaling were the most highly enriched, similar to the yeast SH3 interactome network (Tonikian *et al*, 2009; Supplementary Tables 7 and 8). To compare SH3 domain function in worm and yeast, we compared enriched gene functions among their respective interactomes using a visual map (Figure 4). The map shows that endocytosis and related functions are shared between yeast and worm SH3 interactomes. In addition, the SH3-mediated interactions have yeast-specific roles, such as sporulation, and worm-specific functions, including phagocytosis and multicellular organism development. Thus, the SH3 domain interactome appears to maintain a highly enriched and conserved role in the dynamic process of vesicle-mediated endocytosis over 1.5 billion years of evolution (Wang *et al*, 1999) between the single-cell yeast and the multicellular worm.

## SH3-mediated protein interactions are rewired between yeast and worm

Given the functional similarity between yeast and worm SH3 interactomes, we expected conservation of some underlying SH3-mediated protein interactions. To probe this, we compared our worm SH3 interactome with known protein interactions in yeast (see Materials and methods). In total, 37 SH3-mediated worm PPIs occur between the 98 worm proteins with yeast orthologs (10 SH3-containing proteins and 90 preys, with two baits also found as preys) and 61 yeast PPIs occur between the corresponding yeast orthologs (Figure 5). Only two of these interactions overlap (Figure 5, red lines), a level of conservation no better than expected by chance ($P > 0.5$, Fisher's exact test, assuming a total number of possible interactions of $90 \times 10 = 900$, see Materials and methods). This feature appears to be specific to the SH3-mediated interactions in our interactome, compared with other interactions we obtained with Y2H when using highly connected preys as baits. Among the non-SH3-mediated interactions present in our network, 15 of them connect worm proteins with yeast orthologs, 7 yeast PPIs are found between these orthologs, and 5 of them are conserved between worm and yeast. This gives rise to a significant conservation ($P = 0.001$), given a total number of possible interactions of

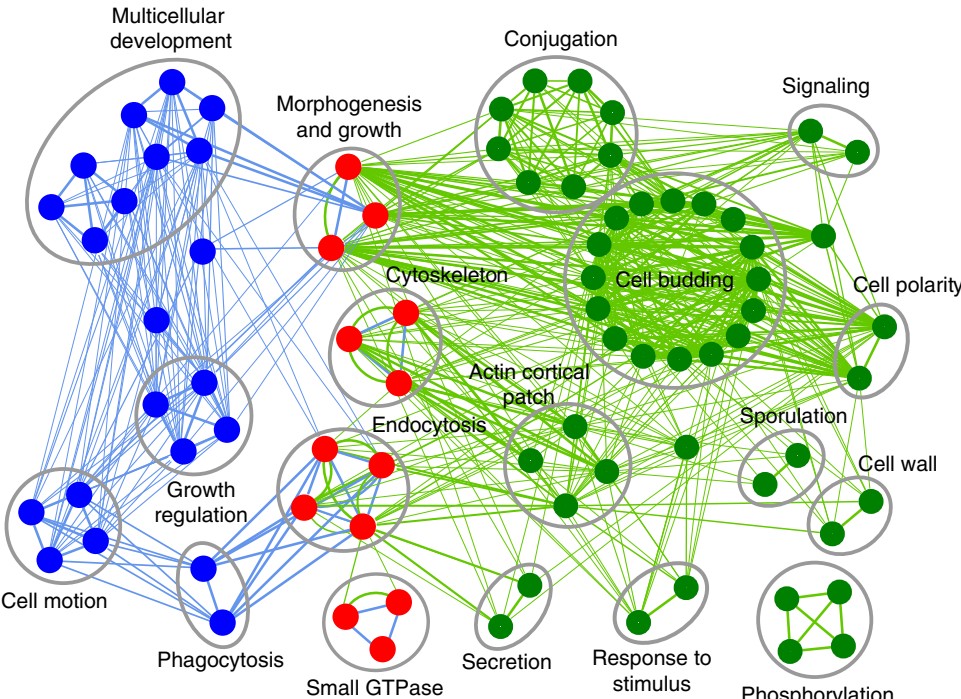

**Figure 4** Endocytosis is conserved between yeast and worm SH3 interactomes and is connected to species-specific functions. Gene functions significantly enriched in yeast and worm SH3 interactomes are visualized as an enrichment map (Merico *et al*, 2010). Nodes represent enriched gene functions in yeast (green), worm (blue), and both organisms (red). Edges link gene functions that share genes. Edge thickness is proportional to number of shared genes and edge color is blue for shared worm genes and green for shared yeast genes. Clusters of functionally related nodes were manually circled and labeled.

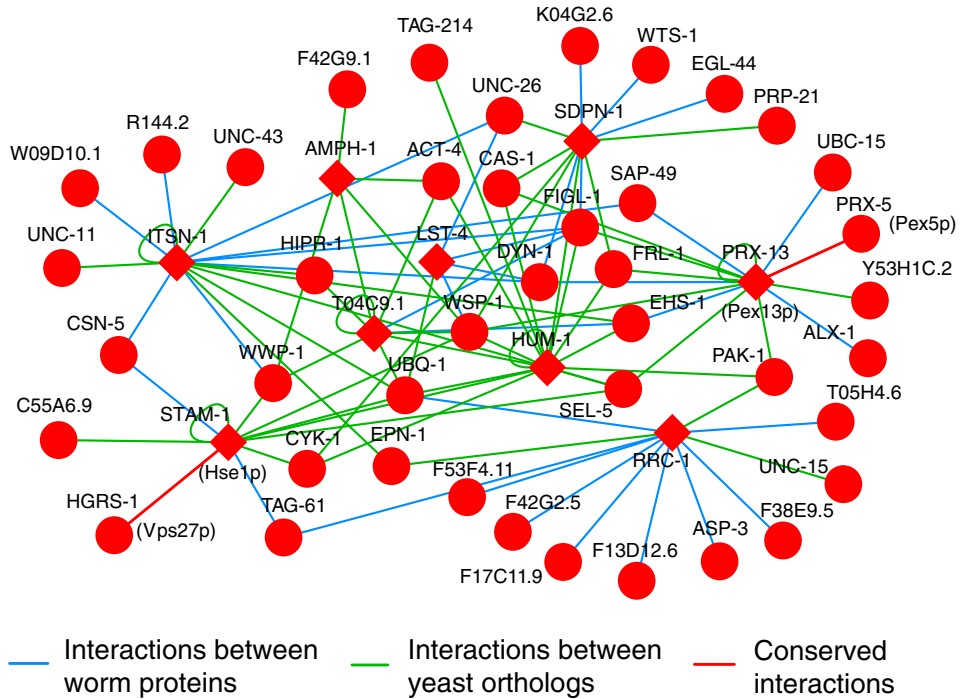

**Figure 5** SH3-mediated PPIs are poorly conserved from yeast to worm. PPIs between worm proteins from our network with yeast orthologs are shown (worm PPIs are blue, yeast PPIs are green, and conserved PPIs are red). Diamonds indicate SH3 containing baits. Names in parenthesis indicate yeast orthologs of worm proteins involved in conserved interactions. Conserved worm proteins that are not involved in interactions with other conserved worm proteins are not shown.

87 (4 conserved baits and 23 conserved preys, with 1 bait also found as a prey). Further, the low yeast-worm conservation of SH3-mediated interactions is not observed when comparing worm and human interactomes. In total, 536 worm PPIs from our interactome involve worm proteins with human orthologs, which participate in 265 published human PPIs compiled by

the BioGRID database (Breitkreutz *et al*, 2008), 31 of which are conserved (Supplementary Figure 3). This overlap is statistically significant ($P = 1.5 \times 10^{-7}$, Fisher's exact test), and likely underestimates the true number of conserved interactions, as no comprehensive human SH3-mediated PPI network is available. Moreover, at least 14 worm SH3 domains have human orthologs with highly conserved SH3 domain sequences ($>55\%$ sequence identity), suggesting that their specificity is also conserved (see Supplementary Figure 1). Thus, our results suggest that extensive SH3-mediated PPI rewiring has occurred between multicellular worm and unicellular yeast.

## Rewiring scenarios

The rewiring observed between worm and yeast may be due to a lack of conservation in: (i) the SH3 domain specificity (or the SH3 domain presence); (ii) the ligand; or (iii) both. For SH3-mediated interactions whose ligand binding motif can be predicted using the phage-derived binding specificity, our data enable us to estimate the frequency of these different rewiring scenarios. We analyzed all SH3-mediated interactions present in either our worm SH3 interactome or the yeast SH3 interactome of Tonikian *et al* (2007), but not both (also excluding interactions found to be conserved by considering published data in the BioGRID database), and for which the binding motifs can be predicted based on our SH3 binding profiles (see Materials and methods and Supplementary Table 9). We searched for the most similar sequence motifs in the ortholog proteins to quantify the motif conservation. Conserved motifs were defined using a sequence identity threshold of 50%. The first rewiring case occurred 11 times (see Supplementary Table 9). For instance, the yeast Myo5p SH3 domain (recognizing PXXXPP, see Figure 2) binds the yeast Vrp1p protein. Although the proline-rich region of Vrp1p is conserved in the worm ortholog WIP-1, the specificity of the worm ortholog of Myo5p, HUM-1 (recognizing PXXPXK or RXLPXXP), is not conserved and WIP-1 and HUM-1 were not observed to interact in the worm SH3 network (see Supplementary Figure 4). The second rewiring case occurred 30 times and is illustrated by SDPN-1 and its yeast ortholog Bzz1p—both display similar SH3 specificity (Figure 2), but none of their greater than 20 respective PPIs are conserved. The low binding site conservation is evident by observing, for instance, that Bzz1p SH3 domain specificity matches a motif in Yta12p (RNIPPPPP, residues 152–159) (Tonikian *et al*, 2009), which is not found in the worm ortholog YMEL-1, despite an overall high protein sequence similarity (62% similarity by BLAST; Supplementary Figure 4). The third case also occurred frequently (63 times in total). Finally, a number of non-conserved interactions could not be reliably mapped at the binding site level because of the absence of specificity profiles for the SH3 domain or the low PWM scores of the predicted motifs (see Supplementary Table 9).

## Domain architecture and binding motifs evolve to rewire PPIs

We asked how evolutionary mechanisms at the protein sequence level involving SH3 domains and their peptide ligands are involved in the rewiring we observe at the PPI level between yeast and worm. There are approximately three times more SH3 domains and SH3-containing proteins in worm compared with yeast. The domain composition of worm SH3 proteins is also more complex than that of yeast (Supplementary Figure 5; Supplementary information). Worm SH3 proteins carry slightly more SH3 and substantially more non-SH3 domains than yeast SH3 proteins (average 1.33 SH3 and 3.56 non-SH3 in worm and 1.17 SH3 and 0.74 non-SH3 in yeast). This indicates that SH3 domains present in the common ancestor of yeast and worm were copied and shuffled to form new worm proteins, likely with new functions, consistent with previous domain architecture evolution studies (Karev *et al*, 2002; Vogel *et al*, 2004; Jin *et al*, 2009).

We reasoned that these evolutionary processes may be responsible for creating novel connections, via evolution of new proteins and PPIs connecting endocytosis to other functions. We manually compiled a comprehensive list of known worm ($n = 109$) endocytosis proteins from the literature (Supplementary Table 10) and known yeast ($n = 85$) endocytosis protein from GO (Ashburner *et al*, 2000) using only experimental evidence codes (Supplementary Table 11). Forty-two worm proteins from our SH3 interactome are present on this list and ten have conserved endocytosis function in yeast. Thus, the majority of worm endocytosis proteins in our network are novel compared with yeast. Eleven of these novel endocytosis proteins directly interact with at least one of the ten functionally conserved worm endocytosis proteins and seven others directly bind one of these eleven proteins. This suggests that many novel worm endocytosis proteins have been recruited by conserved endocytosis proteins. This recruitment can be mediated by conserved endocytosis SH3-containing proteins that bind to species-specific endocytosis proteins, or conversely by functionally conserved proteins binding to species-specific SH3-containing proteins. The former case is observed eight times in our network and involves SH3-containing proteins such as ITSN-1 or SDPN-1. The latter case is seen seven times, involving the four SH3-containing proteins CED-2, NCK-1, TOCA-1, and UNC-57. More generally, of the 34 worm SH3-mediated PPIs involving two endocytosis proteins, only two consist of proteins with both yeast orthologs involved in endocytosis: ITSN-1 (Ede1p in yeast) and SDPN-1 (Bzz1p in yeast) interacting with DYN-1 (Vps1p in yeast—interactions are not known to take place in yeast). The other interactions include worm protein pairs with either one ($n = 15 = 8 + 7$) or both ($n = 17$) yeast orthologs not known to act in endocytosis, or not conserved. These observations support a model where some core endocytosis proteins, often containing SH3 domains, are functionally conserved and are used to recruit other proteins in an organism-specific manner.

To further examine functional connectivity within endocytosis in worm, we classified the 109 worm endocytosis proteins into five major sub-categories: endocytosis, endosome, synaptic vesicle trafficking, phagocytosis, and cytoskeleton (Supplementary Table 10). We observe that worm SH3-mediated PPIs are often found between gene function sub-categories (only 28% of them link proteins of the same sub-category, $P \approx 0.28$), whereas published PPIs among known endocytosis proteins, excluding SH3 proteins, are

clearly enriched in connections within the same gene function sub-category (57%, $P = 0.004$) (Supplementary Figure 6). For instance, endocytosis proteins interact with proteins in the four other major function sub-categories through 17 SH3-mediated PPIs and only 9 non-SH3 PPIs (see Supplementary information). This analysis considers 57 PPIs mediated by individual SH3 domains with 41 of them (72%) linking different sub-categories, and 28 PPIs mediated by non-SH3 proteins with 12 of them (43%) linking proteins in different sub-categories (Supplementary Table 12). Thus, SH3 proteins often function as adapters between functions. Together, these results suggest that creation of SH3 proteins and binding motifs has served to link endocytosis to new pathways over evolution.

## New binding motifs can prevent excessive competition between SH3 domains

The presence of a higher number of SH3 domains in worm could lead to increased ligand competition, which may pressure ligands to adapt. To study this, we compared the number of SH3 domains or predicted binding motifs in a protein with its number of protein interactions in the subset of our worm SH3 interactome with available binding motif information (see Materials and methods). We observe a clear correlation between the number of predicted SH3 binding motifs on a protein and the number of different SH3 proteins binding to it, in both the worm and yeast SH3 interactome (worm: Spearman $\rho^2 = 0.32$, $P = 9.3 \times 10^{-16}$, Figure 6A; yeast:

Spearman $\rho^2 = 0.28$, $P = 3.4 \times 10^{-22}$, Figure 6B). This suggests that, on average, proteins targeted by many SH3 domains have evolved several binding motifs to enable the interacting SH3 domains to coincidentally bind these proteins and prevent excessively competitive interactions among SH3 domains (Zarrinpar *et al*, 2003b). One example of such a potentially coincident PPI involves ABI-1 SH3#1 and NCK-1 SH3#3, which both bind to B0303.7 with different specificities at two distinct binding regions: PPPTPPRKPNI (amino-acid residues 5–15) for ABI-1#1 and PPVPRH (residues 174–179) for NCK-1#3 (Supplementary Figure 7A). ABI-1 and NCK-1 are known to function together in the regulation of actin dynamics (Schmidt *et al*, 2009) and our analysis suggests that B0303.7 could act as an adapter in this pathway. Second, there is no correlation between the number of SH3 domains in a protein and the number of interactions it participates in (worm: Spearman $\rho^2 = 0.13$, $P = 0.005$, Figure 6C; yeast: Spearman $\rho^2 = 0.023$, $P = 0.5$, Figure 6D). For instance, worm SRC-1 has one SH3 domain and almost 80 PPIs, while ITSN-1 has five SH3 domains and < 20 PPIs. Thus, the addition of new domains to a protein does not necessarily lead to new PPIs, but the addition of new motifs often does.

## The worm SH3 interactome identifies mechanism of action for endocytosis proteins

To demonstrate the utility of our worm SH3 data for elucidating cellular function, we focused on AMPH-1, a *C. elegans* BAR and SH3 domain protein homologous to

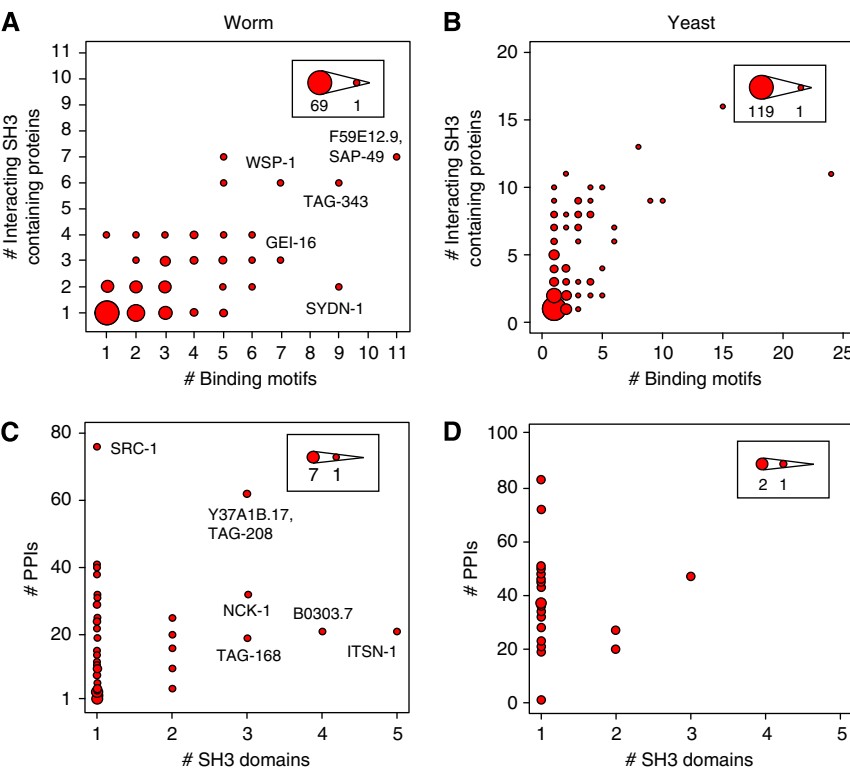

**Figure 6** Binding motif distribution is optimized to prevent excessively competitive interactions. (**A**, **B**) Number of binding motifs is correlated with the number of interacting SH3 proteins, in the subset of our worm (A) and yeast (B) interactomes with predicted binding sites. (**C**, **D**) No correlation is observed between number of SH3 domains in a protein and its number of PPIs in the same worm (C) and yeast (D) SH3 interactomes as in (A) and (B). Circle area is proportional to the number of proteins represented.

mammalian Amphiphysin 2 (also known as BIN1). Both *C. elegans* AMPH-1 and human Amphiphysin 2 localize to recycling endosomes *in vivo* and are required for recycling endosome function (Pant *et al*, 2009). The BAR domain of AMPH-1 binds directly to membranes (Pant *et al*, 2009) and the SH3 domain is presumed to bind to proteins that function with AMPH-1 in membrane trafficking. AMPH-1 did not result in confident Y2H hits; thus, we used the phage-derived PWM to predict AMPH-1 SH3 domain interactors. Among the top scoring candidates was TBC-2, which was recently identified as a GTPase activating protein (GAP) for RAB-5, a master regulator of endocytosis and the early endosome (Chotard *et al*, 2010; Li *et al*, 2009). Without TBC-2, RAB-5 fails to be properly downregulated and endosome function is compromised (Chotard *et al*, 2010). While TBC-2 is known to localize to endosomes and to be important for endosome function, the mechanism of its recruitment to endosomes was poorly understood. We confirmed the physical interaction between the AMPH-1 SH3 domain and TBC-2 via GST pulldown of *in vitro* translated HA-tagged TBC-2 (Figure 7A). This interaction is specific, as the SH3 domain of SDPN-1 failed to bind to TBC-2 in the assay (Figure 7A). To determine if the predicted AMPH-1 interaction is important for TBC-2 recruitment to endosomes, we examined the subcellular localization of a functional GFP-TBC-2 fusion protein in living *C. elegans*. In agreement with previously published work, we found that in a wild-type background GFP-TBC-2 was localized to numerous intracellular puncta previously identified as endosomes (Chotard *et al*, 2010; Figure 7B and E). However,

GFP-TBC-2 was diffusely localized in an *amph-1* deletion mutant background, indicating that AMPH-1 is required *in vivo* for efficient recruitment of TBC-2 to endosomal membranes (Figure 7C and E). This is likely specific, as previous work showed that loss of AMPH-1 does not result in a large-scale redistribution of endosome proteins to the cytoplasm, but rather is likely to only occur for proteins directly interacting with AMPH-1 (Pant *et al*, 2009). Furthermore, we found that loss of the known AMPH-1 partner protein RME-1 also disrupted the localization of TBC-2 to endosomes (Figure 7D and E), suggesting that the intact AMPH-1/RME-1 complex is required for endosomal recruitment of TBC-2.

## The worm SH3 interactome identifies new endocytosis proteins in worm and human

Proteins of unknown function interacting with endocytosis proteins are likely involved in related processes, based on the concept of guilt-by-association. We predicted 86 new worm endocytosis proteins that are highly connected in our interactome to those on our manually curated list of known endocytosis proteins (Supplementary Table 10) using a guilt-by-association strategy based on a modified k-core algorithm (see Materials and methods and Supplementary Table 13). This approach yields reasonable prediction performance (verified using 10-fold cross-validation, area under the receiver operating characteristic (ROC) curve (AUC) of 0.69, $P = 10^{-6}$).

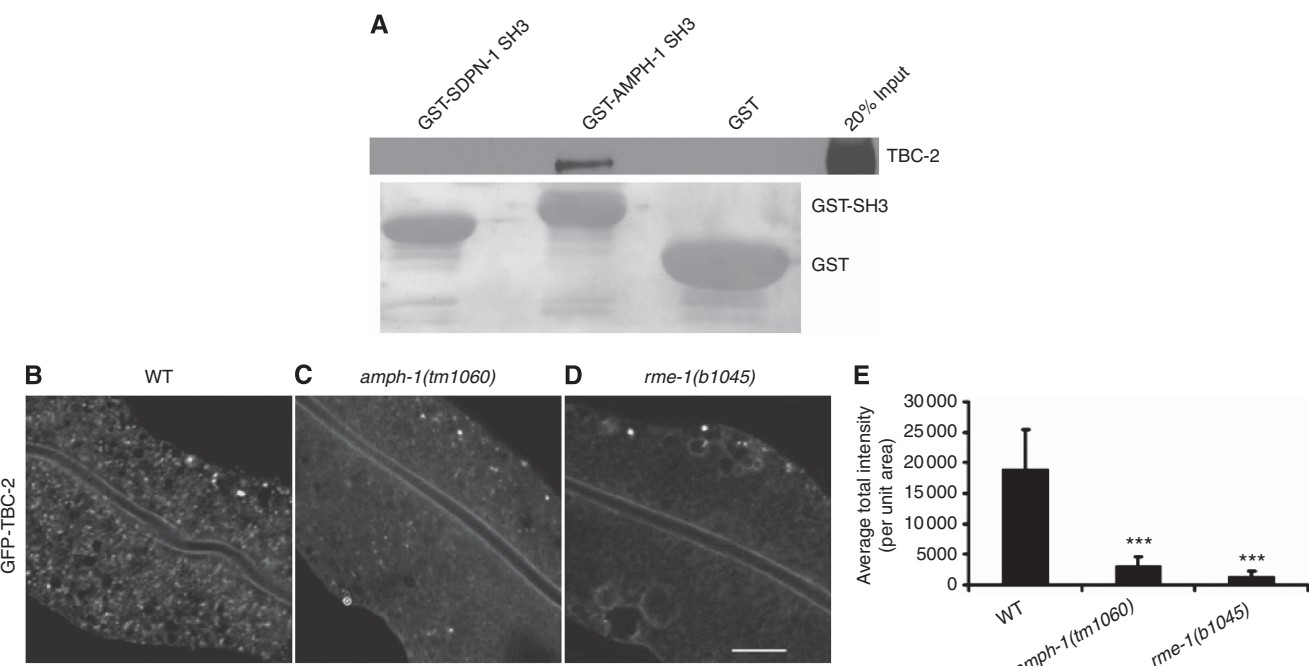

**Figure 7** Experimental validation of predicted interactions and endocytosis functions in worm. (**A**) The worm AMPH-1 SH3 domain physically interacts with TBC-2 and not the SDPN-1 SH3 domain or GST controls, as shown by western blot. GST bait proteins, visualized by Ponceau S staining, are shown under the corresponding western blot lanes. (**B–D**) GFP-TBC-2 localization to endosomes (B) is disrupted in amph-1 (C) and rme-1 (D) mutants. Comparable confocal images show living intestinal epithelial cells expressing GFP-tagged TBC-2 from an integrated low-copy number transgene in wild-type, *amph-1(tm1060)*, and *rme-1(b1045)* mutant animals. Scale bar, 10 μm. (**E**) Average fluorescence intensity of GFP-TBC-2 labeled puncta quantifies disruption of TBC-2 endosome localization across multiple experiments. Error bars, standard deviation (*n* = 18 each, six animals of each genotype sampled in three different regions of each intestine). Significant differences in the one-tailed Student's *t*-test are indicated (\*\*\**P* = 0.001).

The observed protein interaction conservation between human and worm suggests that our predicted endocytosis functions for worm proteins are transferable to human proteins by orthology. Of our 86 predicted novel worm endocytosis proteins, 55 have human orthologs, of which at least 10 are known to function in various endocytosis pathways in human or other mammals, including SRC (Cao *et al*, 2010), MAP3K10 (Akbarzadeh *et al*, 2002), NOSTRIN (Icking *et al*, 2005), and SNX18/SNX9 (Park *et al*, 2010; Supplementary Table 13). To test if these human orthologs are involved in endocytosis, we co-localized 30 of them with clathrin light chain A (CLTA), a marker of the major internalization pathway—clathrin-mediated endocytosis, in a human melanoma cell line. Four proteins co-localized with CLTA, suggesting their endocytosis role in human cells, of which two (SH3D19 and MAP3K9) were previously unreported (Figure 8A–F; Supplementary Figure 8). Another 14 were localized to endocytosis-linked structures, including focal adhesion-like structures, actin fibre-like structures, membrane ruffles, cell surface puncta or tubules, while the rest were cytoplasmic (Figure 8G–I; Supplementary Table 14; Supplementary Figures 9 and 10). As an additional experimental test of our predictions, we compared our results with an available data set derived from light endosome, heavy endosome, and lysosome fractions (McCaffrey *et al*, 2009) from human neuroblastoma cell lines that used

phospho-proteomics to identify over 1100 phosphotyrosine-containing proteins in these fractions (see Materials and methods). Our predicted human endocytosis proteins significantly overlap this list ($n = 28$, $P = 7.9 \times 10^{-9}$, Fisher's exact test; Supplementary Table 15). Thus, our worm SH3 domain interactome is useful for predicting endocytosis and related functions in worm and human.

## Discussion

Protein interactions mediated by PRMs encode key parts of the wiring diagram of cellular signaling pathways. In this work, we studied the SH3 domain interactome in *C. elegans*, following up on previous work mapping the equivalent network for budding yeast (Tong *et al*, 2002; Tonikian *et al*, 2009). We surveyed the repertoire of worm SH3 binding specificity using billions of peptides in phage display assays. We then mapped the worm SH3 interactome using large-scale stringent Y2H screens. Combining these two data sets, we created a high-confidence SH3 domain interactome containing 1070 PPIs connecting 79 worm SH3 domains and 475 proteins, with detailed binding site information. Twenty five of the sixty-three SH3-containing proteins that we mapped interactions for have never been reported to interact with any protein, indicating that a substantial fraction of our worm SH3

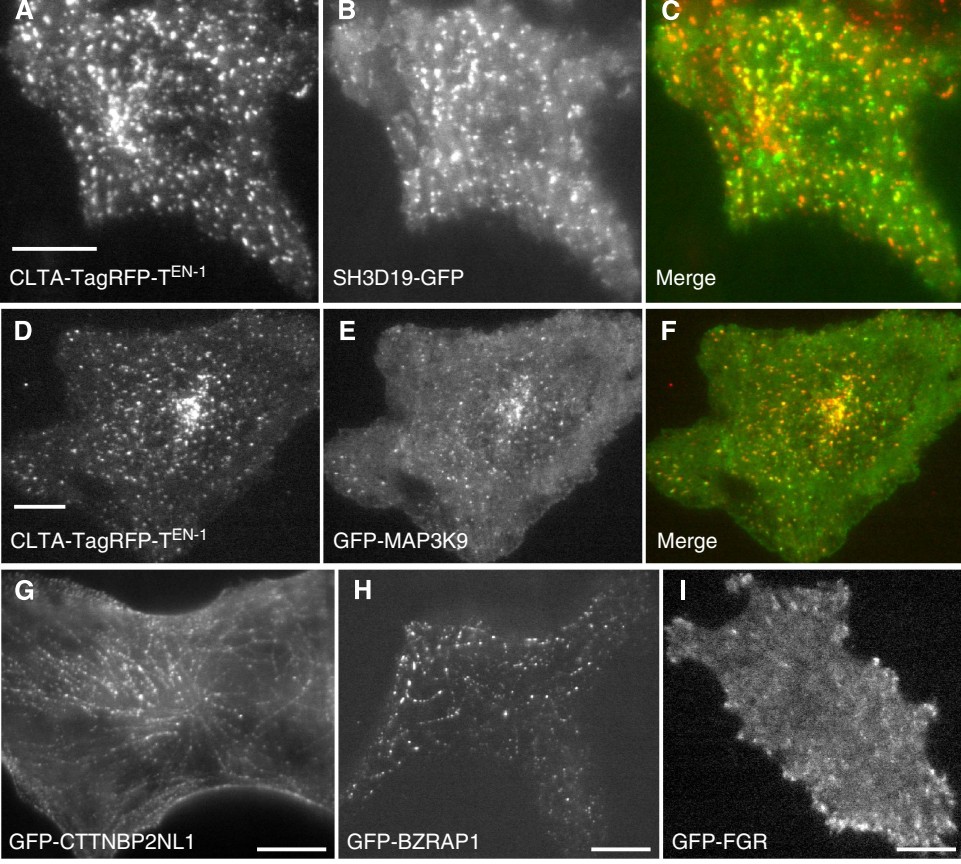

**Figure 8** Experimental validation of predicted endocytosis functions in human. (**A–C**) Co-localization of SH3D19-GFP (C-terminal tag) with CLTA-TagRFP-T[EN-1] in SK-MEL-2 cells. (**D–F**) Co-localization of GFP-MAP3K9 (N-terminal tag) with CLTA-TagRFP-T[EN-1]. (**G, H**) Localization of GFP-CTTNBP2NL and GFP-BZRAP1 to actin stress fibre-like structures. (**I**) Localization of GFP-FGR to focal adhesion-like structures. Scale bars, 10 μm.

interactome is novel. Comparing our new map with the equivalent map for budding yeast *S. cerevisiae*, we discovered that the SH3 domain family specificity repertoire and endocytosis function are conserved between yeast and worm over 1.5 billion years of evolution, but strikingly, the single domain specificity is only partly conserved and the actual wiring of the network is hardly conserved at all. Further, our protein interaction data predicted novel aspects of endocytosis in worm and human, which we experimentally validated.

The evolution of molecular interactions and its implication on protein function conservation have been increasingly studied over the past decade. For instance, interologs were proposed as a method to transfer PPIs between species for orthologous interacting protein pairs (Matthews *et al*, 2001). This is useful for information transfer and gene function prediction, and may be reasonable for well-conserved complexes (Brown and Jurisica, 2007; van Dam and Snel, 2008), modules (Zinman *et al*, 2011), and paralogs (Mika and Rost, 2006). However, frequent network rewiring has often been observed in PPI and transcriptional regulatory networks (Ihmels *et al*, 2005; Beltrao and Serrano, 2007; Wang *et al*, 2010; Kim *et al*, 2012) and multiple studies have experimentally shown that extensive rewiring is tolerated in real biological pathways (Isalan *et al*, 2008; Mody *et al*, 2009; Peisajovich *et al*, 2010; Xie *et al*, 2010). This is especially true for interactions involving disordered regions, such as those mediated by SH3 domains (Mosca *et al*, 2012; Sun *et al*, 2012). Studying network evolution mechanisms is crucial to better understand how protein function evolves and more accurately transfer it across organisms. Early theoretical network evolution models include the birth, death and innovation of protein domains model, which predicts a PPI network with similar topological properties to known networks (Karev *et al*, 2002) and a gene duplication, network divergence model (Wagner, 2003) that has been observed to occur in real networks (e.g., *Arabidopsis* Interactome Mapping Consortium, 2011). The gene duplication, network divergence model has also been extended to protein complexes (Yosef *et al*, 2009) and network motifs (Ward and Thornton, 2007). More detailed work has examined how molecular binding site changes have rewired interactions, in particular noting that short DNA and protein sequence motifs are easier to evolve than larger binding sites (Neduva and Russell, 2005) and multiple studies have examined specific mechanisms of short linear motif evolution (Moses *et al*, 2007; Tan *et al*, 2009; Freschi *et al*, 2011; Sun *et al*, 2012; Habib *et al*, 2012). Our work is the first, to our knowledge, to investigate protein interaction network evolution between unicellular and multicellular eukaryotes at binding site resolution on a large scale. We show how a protein interaction network can substantially rewire and 'recruit' conserved functions, like endocytosis, for use in organism-specific processes, such as sporulation in yeast and phagocytosis in worm but can also maintain certain aspects of its function. We also propose a mechanistic model involving small linear motifs evolving to create new interactions linking biological processes together. Our results agree with previous work, in particular observing that TF networks are frequently rewired, but often maintain their biological process function (Habib *et al*, 2012), suggesting similarities of evolutionary mechanism in diverse networks involving short linear motifs.

We also observe less network conservation between yeast and worm than between worm and human, suggesting that similarly to biological sequences, the network evolution rate is variable over time—variability that also depends on network structure (Brown and Jurisica, 2007) and biological process (Beltrao and Serrano, 2007). Continued work will ideally converge on a general theory of network evolution that will include accurate measurements of network evolutionary rates and lead to more accurate function transfer between genes and proteins.

Since any large-scale PPI mapping methodology results in some false positives and false negatives, it may be argued that the low conservation we observe between yeast and worm might simply be due to noisy data. This is unlikely for two main reasons. First, phage display and stringent Y2H are orthogonal in their respective strengths and weaknesses. Phage display uses *in vitro* binding and short synthetic peptides, whereas two-hybrid analysis uses *in vivo* binding and native proteins or protein domains. Nevertheless, we observe significant agreement between the SH3 domain interaction network results derived from these different techniques (Tong *et al*, 2002; Tonikian *et al*, 2009). Second, we observe significant PPI conservation between worm and human that cannot be explained by chance (Supplementary Figure 3). Our *C. elegans* SH3 domain interaction network, if noisy, could not simultaneously show poor conservation with yeast and higher conservation with human. Thus, our data clearly indicate that SH3 domain interactions in metazoans are frequently rewired and fundamentally different from that in the unicellular yeast.

In a system like the SH3 interactome that involves many domains that recognize many similar peptide ligands, it is important that evolution avoid creating competition among components, which could lead to pathways interfering with one another (Zarrinpar *et al*, 2003b). We studied this in individual proteins that contain one or more SH3 domains or binding motifs, as all domains and motifs within a protein are co-expressed and co-localized. We did not observe any correlation between the number of SH3 domains in a protein and the number of interacting partners. From this, it appears that multiple domains on the same protein have not been widely used over evolution to avoid competitive interactions. We suggest, based on functional analysis (Supplementary Figure 6), that this enables the proteins to act as scaffolds or adapters between different biological systems (Pawson and Nash, 2003). However, we found a clear correlation between the number of binding motifs and the number of interacting SH3-containing proteins. This suggests that novel instances of linear motifs evolved to enable multiple SH3 domains to target different regions of the same interacting partners without competing with one another. While reduced competition among SH3 domains is a general trend in our data, we still observe instances where two SH3 domains target the same binding site (Supplementary Table 16). Some of these cases may not be relevant *in vivo* due to different expression or subcellular localization that would prevent competition, but some cases may occur and be beneficial, for instance, to mediate switches in regulatory processes. One such situation arises with the SH3 domains of endophilin (UNC-57) and syndapin (SDPN-1). Both domains are predicted to bind to the

proline-rich motif PRGGPGAPPPPGMRP (residues 783–797) of dynamin (DYN-1) (Supplementary Figure 7B). This is consistent with the known mutually exclusive interactions between the rat dynamin and syndapin or endophilin proteins (Anggono and Robinson, 2007). Thus, we expect worm syndapin and endophilin to function either in the same stage during endocytosis with a competitive regulatory relationship, or in different stages or locations during synaptic vesicle endocytosis (Anggono and Robinson, 2007).

We showed how the worm SH3 interactome can be used to identify mechanism of action for proteins known to localize to endocytosis-related structures and also can be used to identify new protein participants of endocytosis and related pathways. Our predicted endocytosis proteins co-localized with markers of endocytosis pathways or localized to potentially related structures, including focal adhesion-like and actin fibre-like structures. While we did not precisely identify these related structures using markers, their presence agrees with our model of SH3 proteins acting as adapters linking endocytosis to related important signaling pathways and structures.

The low PPI conservation between worm and yeast SH3 interactomes has important implications for predicting protein function by homology. We can successfully link the SH3 domain family to endocytosis by studying yeast, but the details of the system cannot be extrapolated to higher organisms. However, we observe higher conservation between worm and human compared with yeast for both SH3 domains and their interactions (see Supplementary Figure 3 and Materials and methods), suggesting that the worm SH3 interactome may be at least partly predictive of human PPIs. It will be important to map additional interactomes at binding site resolution across species to gain further insight into the generality of our observation that function is conserved over form from unicellular to multicellular eukaryotes.

# Materials and methods

## Experimental procedures

### *C. elegans* SH3 domain identification
*C. elegans* SH3 domain boundaries were defined as the union of the domain regions identified by CDD (Marchler-Bauer *et al*, 2009), Pfam (Finn *et al*, 2008), and SMART (Schultz *et al*, 2000; Letunic *et al*, 2009) based on WormBase Release WS100 (Schwarz *et al*, 2006). For cloning, an additional 10 amino acids on either side were added (where applicable) (Supplementary Table 1).

### SH3 domain GST fusion plasmid construction
DNA fragments encoding the identified domains were amplified from a *C. elegans* cDNA library (custom library purchased from DualSystems) by PCR and cloned into a vector designed for the expression and purification of SH3 domains fused to the C-terminus of glutathione S-transferase, as previously described (Tonikian *et al*, 2007). All plasmid constructs were verified by DNA sequencing.

### Phage display selection of peptide ligands
Phage-displayed peptide libraries ($>10^{10}$ unique members) fused to the N-terminus of the gene-8 major coat protein of M13 filamentous phage were used to select peptide ligands for the collection of purified GST-SH3 fusion proteins. All domains were first screened using a random dodecapeptide library ($X_{12}$, where X is any amino acid). Domains that failed to select peptides with the dodecapeptide library

were subsequently screened using a biased peptide library ($X_6$-PXXP-$X_6$, where P is proline, (*) in Figure 1), or both the $X_{12}$ and $X_6$-PXXP-$X_6$ libraries ((**) in Figure 1). Certain yeast SH3 domains, annotated with (***) in Figure 1, were also tested using a biased library containing a fixed positive charge (X7-R/K-X7, where R and K are arginine and lysine, respectively), as previously published (Tonikian *et al*, 2009). Phage display selections were carried out as previously described (Tonikian *et al*, 2007). Individual binding clones were tested for positive interactions with cognate worm SH3 domains by phage ELISA as previously described (Tonikian *et al*, 2007). The sequencing of $\sim 3000$ clones resulted in the isolation of 1330 unique peptide sequences in total for 36 SH3 domains. All phage results are available in Supplementary Table 17 (see also http://baderlab.org/Data/SH3Worm for alternative data formats).

## Y2H plasmid construction using homologous recombination cloning
DNA fragments encoding SH3 domains and other baits were amplified by PCR from a *C. elegans* cDNA library (DualSystems), using sequence-specific primers fused to common sequences designed for homologous recombination cloning as previously described (Tonikian *et al*, 2009). All plasmid constructs were verified by DNA sequencing.

## ORFeome Y2H screening
All bait plasmids were transformed into Mav203 (Li *et al*, 2004) and Y8930 (Yu *et al*, 2008) yeast strains. *C. elegans* AD-ORFeome v1.1 (Reboul *et al*, 2003) and v3.1 (Lamesch *et al*, 2004), which contain about 13 000 AD-ORFs in total, were transformed into MaV103 (Li *et al*, 2004) and Y8800 (same genotype as Y8930 except opposite mating type) in this study. ORFeome Y2H screening was performed using a mating method, as previously described (Rual *et al*, 2005), with different yeast strain combinations, that is, Y8930 × Y8800, MaV203 × MaV103, Y8930 × MaV103, and MaV203 × Y8800. SD-Leu-Trp-His + 3-AT (3-amino-1,2,4-triazole) selective medium was used for screening. The optimal concentration of 3-AT for each screen was chosen using the diploid strain generated from the mating of bait yeast carrying bait plasmid with the prey yeast carrying empty prey vector. Plates were incubated at 30°C for 5–10 days before scoring and picking positive colonies. All positive colonies passing stringent interaction confirmation were identified by colony-PCR and DNA sequencing as previously described (Rual *et al*, 2005). Further, we previously screened 12 out of the 80 worm SH3 domains as baits using two Shifted Transversal Design (STD) pooling designs, one-on-one array-based Y2H and screening-sequencing methods (Xin *et al*, 2009). This data set is included in the worm SH3 domain interactome presented here. Altogether, we identified 731 unique domain–protein interactions involving 79 SH3 domains (in 59 SH3 proteins) and 313 interactors from ORFeome Y2H.

## cDNA library Y2H screening
A custom mixed-stage *C. elegans* cDNA library (DualSystems) with a complexity of $2.5 \times 10^7$ and insert average size of 1.5 kb was transformed into Y8800 (Yu *et al*, 2008). Screening was performed by mating methods as previously described (Kolonin *et al*, 2000) with some modifications. For each screen, $>5 \times 10^7$ diploid colonies were screened on SD-Leu-Trp-His + 3AT plates, and up to 200 positive single colonies (where applicable) were picked. Similar to the ORFeome Y2H, all positive colonies passing a stringent phenotyping assay were identified by colony-PCR and DNA sequencing as previously described (Rual *et al*, 2005). From about 10 000 sequencing results from our cDNA library Y2H, we identified 934 unique domain–protein interactions between 69 SH3 domains (in 51 SH3 proteins) and 463 interactors.

## *C. elegans* endosome analysis
Strain construction was performed by standard methods and mutant genotype was confirmed by PCR analysis. In Figure 7A, Glutathione beads loaded with recombinant GST, GST-AMPH-1(SH3),

or GST-SDPN-1(SH3) were incubated with *in vitro*-expressed HA-tagged TBC-2, and then washed to remove unbound proteins. Bound proteins were eluted and analyzed by western blot using anti-HA antibodies. In Figure 7B–D, transgenic GFP-TBC-2 expressing worms (Chotard *et al*, 2010) were a generous gift from Dr Christian Rocheleau (McGill University). Live worms were mounted on 2% agarose pads containing 100 mM tetramisole (MP Biomedicals, OH) in M9 buffer. Young adult hermaphrodites grown 24 h past the L4 larval stage were used for imaging. To obtain images of GFP fluorescence without interference from autofluorescence, the spectral fingerprinting function of a Zeiss LSM510 Meta confocal microscope system (Carl Zeiss MicroImaging) was used, as previously described (Chen *et al*, 2006a). Figure 7E displays the quantification of confocal images, which was performed using MetaMorph software version 6.3r2. The same threshold values were used for all images within a given experiment. For each marker comparison, at least six animals were analyzed. Three randomly selected regions per animal were analyzed using circular regions of defined area. Quantification of fluorescence intensities or object count was performed. Student's *t*-test was used to determine the difference between the different groups (Figure 7).

### Protein localization assay in human cell lines

Human skin melanoma SK-MEL-2 CLTA-TagRFP-T gene-edited cells (SK-MEL-2 CLTA[EN-1]) (Doyon *et al*, 2011) were maintained under 5% $CO_2$ at 37°C in DMEM/F-12 (Invitrogen) supplemented with 10% FBS (fetal bovine serum, HyClone). SK-MEL-2 CLTA[EN-1] cells were transiently transfected with plasmids encoding GFP-tagged candidate clones (all N-terminally tagged except for SH3D19, which was C-terminally tagged) using Lipofectamine 2000 according to manufacturer's protocol. Total internal reflection fluorescence (TIRF) microscopy images were captured using MetaMorph software on an Olympus IX-81 microscope using a ×60 NA 1.49 objective and an ORCA-R2 camera (Hamamatsu). The system was maintained at 37°C using a WeatherStation chamber and temperature controller (PrecisionControl). At 16–24 h before imaging, cells were seeded onto uncoated glass coverslips in growth medium. During imaging, cells were maintained in DMEM without phenol red that was supplemented with 5% FBS, 10 mM HEPES and 1 × GlutaMAX. A 488-nm solid-state laser (Melles Griot) and a 561-nm diode-pumped solid-state laser (Melles Griot) were used to excite GFP and RFP fluorophores, respectively. Simultaneous two-color TIRF images were obtained using a DV2 image splitter (MAG Biosystems) to separate GFP and RFP signals. Image sequences were captured at a 2-s per frame interval. Image and kymograph analyses were performed using ImageJ software.

### Human cell endosome phosphoproteomics

Fractionation of endosomes from human neuroblastoma cell lines (SH-SY5Y, SMS-KCN, SK-N-BE(2), and LAN-6) was performed as described in McCaffrey *et al* (2009) with modifications. Briefly, cells were mechanically permeabilized in a cytoplasm-like buffer, and organelles that emerged from permeabilized cells were recovered in the supernatant of 1000 *g*, 10 min centrifugation. Organelles were resolved by mass-by-velocity sedimentation, then density-by-flotation equilibrium centrifugation, on iodixanol gradients. Fractions previously defined as containing lysosomes (lys, ρ = 1.11–1.16 g/ml) and heavy and light endosomes (end1, end2, ρ = 1.08–1.14 g/ml) were pooled from four or more experiments and subjected to PhosphoScan (Cell Signaling Technologies, Danvers, MA) using anti-phosphotyrosine, or anti-Akt-substrate antibodies. Peptides were identified by mass spectrometry as described (Rush *et al*, 2005; Guo *et al*, 2008).

In total, 1139 human proteins were identified with this approach. We compared this data set with the predictions by orthology of human endocytosis proteins based on our worm SH3 interactome. The 55 predicted novel worm endocytosis proteins with human orthologs can be mapped to 143 human proteins. Twenty eight of them overlap with the phosphoproteomics predictions ($P = 7.9 \times 10^{-9}$, assuming a universe of 20 000 proteins, see Supplementary Table 15). All proteins that overlap with predicted human endocytosis list were in the endosome fractions except SFPQ and LASP1 (26 out of 28), which were only found in the lysosome fraction.

## Data analysis

### Building PWMs from phage peptides

Phage display results can be modeled using PWMs (also referred to as Position-Specific Scoring Matrices). To build the PWMs, phage-derived peptides were first aligned manually for each SH3 domain. Two domains (from HUM-1 and STAM-1 proteins) displayed two different kinds of specificity, so the peptides were manually split into two groups accordingly and each group was modeled with a separate PWM (Gfeller *et al*, 2011). Gaps were treated as non-specific amino acids when computing frequencies. A random count was added proportional to the entropy of each column in the alignment. To account for the effect of small fluctuations of non-specific amino acids, the contribution of amino acids whose frequency was not higher than expected by chance ($P > 0.05$) was averaged at each position. Frequencies at each position were normalized to 20, such that a completely non-specific PWM gives a normalized score of 1 for any peptide. Positions on both sides of the central motif in the PWMs with low specificity (using a threshold of 0.76 on the entropy) were not included in the PWMs. PWMs are available in Supplementary Table 18.

### SH3 specificity tree construction

To compute the similarity between PWMs and build the tree displayed in Figure 1, the normalized Euclidean distance $d^{align}$ corresponding to the best alignment between any two PWMs was first computed (here, the best alignment was defined as the one giving the smallest Euclidean distance). For two PWMs with parameters $\theta_{ij}^{(1)}$ and $\theta_{ij}^{(2)}$ corresponding to the probability of observing amino acid $i$ at position $j$, $d^{align}$ is defined as $\left( \frac{\sum_{i,j} \left( \theta_{ij}^{(1)} - \theta_{ij}^{(2)} \right)^2}{2\ell^{(1-2)}} \right)^{1/2}$, where $\ell^{(1-2)}$ stands for the length of the alignment between the two PWMs. Positions on the left or right of a phage-derived PWM that do not overlap with the other one in the alignment were assumed to be random (i.e., all residues were given the same weight). Since non-specific positions tend to display some similarity that is not relevant for comparing PWMs, we further subtracted the Euclidean distance of the two PWMs completely unaligned $d^{unaligned}$ (i.e., with no overlapping positions). The PWM similarity was then defined as $(d^{unaligned} - d^{aligned})/d^{unaligned}$. The tree of Figure 1 was built with this similarity measure, using the average linkage hierarchical clustering algorithm as implemented in R and the graphical layout was generated with TreeDyn (Chevenet *et al*, 2006), followed by manual positioning of sequence logos (Figure 1).

### Computing PWM scores

PWMs can be used to search for sequence motifs matching the specificity of SH3 domains within worm protein sequences. Given a PWM of length $m$, the score of a $m$-mer peptide is simply computed by multiplying the probabilities $\theta_{ij}$ corresponding to the probability of a peptide residues $i$ at each position $j = 1 \ldots m$ in the alignment of phage peptides. The PWM score of a full protein is defined as the score of the highest scoring peptide found by scanning the entire length of the protein. Scores were normalized by $20^{-m}$, such that a completely non-specific PWM gives a score of 1 for any peptide.

### Y2H network filtering

Many PPI mapping technologies are known to have an inherent false positive rate, including Y2H and its variations (Braun *et al*, 2009). To minimize the false positive rate, in addition to the stringent experimental filters described above, we computationally filtered our Y2H PPIs according to the following criteria. First, we excluded Y2H PPIs involving the top four highly connected AD-TFs, namely CEH-17, DPL-1, TTX-1, and VAB-3. These four TFs were identified to interact with 9–38 different baits, while the other AD-TFs only interact with at most two baits (Supplementary Figure 11). Second, all Y2H PPIs identified from at least two independent positive colonies were included in our network. Third, we added Y2H PPIs identified from one positive colony that are found in a published data set (WI8

(Simonis *et al*, 2009), are known interologs (Walhout *et al*, 2000; Matthews *et al*, 2001; Simonis *et al*, 2009), are present in WormNet functional interactions (Lee *et al*, 2008), or are known be functionally related (Supplementary Table 3, column L). Fourth, we included PPIs identified from a single positive colony when a clear sequence motif matched the phage-derived specificity. To determine a reasonable threshold on the scores obtained by scanning the protein sequence with the SH3 PWM, we first observed that the median PWM score distribution of interactions supported by more than one independent positive colony is equal to 972 (see Supplementary Figure 12, red line). Moreover, manual inspection of the predicted binding motifs shows that sequences with scores larger or equal to roughly 1000 clearly show good matches with the SH3 binding specificity. Therefore, a threshold of $T = 1000$ was chosen to include in the network interactions supported by only one yeast colony. These interactions represent only a small fraction of the total network (67 out of 1070). For this reason, all our results are robust to large variations of this threshold. For instance, the number of conserved interactions between worm and yeast or worm and human remains the same for $T = 10$ to $T = 100000$.

As a test of these filtering criteria, we show in Figure 3 that our high confidence network overlaps our phage display results better than the unfiltered Y2H network (i.e., including Y2H interactions supported by only one colony, red curve in Figure 3). The final worm SH3 interactome after combining all Y2H results and filtering as described above contains 1070 PPIs among 91 baits (79 SH3 domains in 63 proteins, 9 full-length hub proteins and 3 ITSN-1 fragments) and 475 interacting proteins. For all interactions involving an SH3 domain with available phage display data, the PWM score, the best scoring peptide as well as the genome-wide rank of this peptide are indicated in Supplementary Table 3. Ranks were computed by comparing the PWM score of the interacting protein with those of all worm proteins (based on the Ensembl 53 release (Hubbard *et al*, 2009), WormBase 180).

## Comparison of Y2H and phage data

We used our SH3 PWMs to score every worm protein in the proteome as a potential SH3 binder, assigning to each the PWM score of the highest scoring amino-acid ligand motif found along its amino-acid sequence (see Supplementary Table 3). Based on these scores, all Y2H PPIs were assigned a genomic rank for each SH3 domain with available phage data (i.e., rank of the SH3 interactor among all PWM scores for the given SH3 domain). To show the enrichment of Y2H PPIs in high PWM scoring proteins, we computed the cumulative distribution of all Y2H PPI ranks, both for the unfiltered Y2H network (all interactions identified in the Y2H screen) and the final filtered SH3 interactome (Figure 3, red and green curves, respectively). For the latter, Y2H interactions included only because of the presence of a motif matching the phage-derived specificity (see above) were not considered, to not bias the statistics. Statistical significance was computed by randomly assigning genomic ranks to each Y2H interaction ($10^7$ randomizations were carried out), using the area under the curve displayed in Figure 3, as a numerical measure of the enrichment. The same analysis was carried out for each domain separately in Supplementary Figure 2. This analysis shows that our filtered network agrees better with our phage data compared with the unfiltered network and the random model. Y2H interactions for which a motif matching the phage specificity could not be identified might result from indirect interactions, non-canonical binding that cannot be detected in phage (e.g., not mediated by short peptides), and false positives inherent to any high-throughput protein interaction detection technique.

## Y2H network quality and coverage evaluation

Four different criteria were used to validate our protein interaction network (Supplementary Tables 4 and 5). First, we compared our data set with the most recent version of the worm protein interactome (WI8) (Simonis *et al*, 2009). Second, we examined the overlap with the worm interolog data set (Simonis *et al*, 2009). Interologs are pairs of interactors that are predicted to physically interact because their respective orthologs interact in another organism (Walhout *et al*, 2000; Yu *et al*, 2004; Simonis *et al*, 2009). Third, we computed the overlap

with WormNet v2 (Lee *et al*, 2008), which is a probabilistic functional gene interaction network built by integrating several sources of evidence, including co-citations, co-expression, and interologs. In each comparison with WI8, interolog, and WormNet, respectively, interactions observed only once in the Y2H screen but included in the SH3 interactome because of their occurrence in the corresponding data set (WI8, WormNet, and interolog, respectively) were not considered. We observed significant overlap in all of the above comparisons (Supplementary Table 4). Fourth, high quality interactomes are expected to be enriched with protein interacting pairs having similar functions (Lord *et al*, 2003; Guo *et al*, 2006). The interacting protein pairs in our SH3 interactome were more likely to share similar functional annotations (GO terms; Ashburner *et al*, 2000) than random data sets (Supplementary Table 5, see below for a more detailed description of this analysis). Taken together, the significant overlap of our worm SH3 interactome with a number of reference data sets indicates its high quality.

Using individual SH3 domains as baits is known to increase interactome mapping coverage (Tong *et al*, 2002; Tonikian *et al*, 2009). Indeed, previous worm interactome mapping efforts using full-length proteins as baits (Li *et al*, 2004; Simonis *et al*, 2009) or protein fragments as baits (Boxem *et al*, 2008) retrieved ~2.3 and ~3.1 interactors per bait on average, respectively, while we retrieved ~12. Further, the WI8 data contain 64 PPIs involving 21 proteins with SH3 domains also present in our network, while our data set contains 457 interactions involving individual SH3 domains from these 21 proteins as baits. We expect that constructing and screening bait collections of SH3 protein fragments, in addition to individual SH3 domains, would further decrease the false negative rate of the Y2H screens.

## GO term semantic similarity analyses

To quantify the functional relatedness between protein pairs, we identified to interact using Y2H as a benchmark for our Y2H data (see above), we used an established semantic similarity scoring scheme based on GO (Lin, 1998). This measure is highly correlated with sequence similarity, expression similarity and protein interactions (Lord *et al*, 2003; Guo *et al*, 2006). We used the 4/25/2009 version of the *C. elegans* GO annotation, which contains 37 313 manually annotated or reviewed, that is, excluding IEA (Inferred from Electronic Annotation) evidence code, biological process annotations for 6341 proteins/genes, 3443 non-IEA cellular component annotations for 1355 proteins/genes, and 6541 non-IEA molecular function annotations for 2523 proteins/genes. Briefly, the GO term semantic similarity score (SIM score) ranges from 0 to 1 where a greater value indicates higher semantic similarity. The score for any two GO terms is calculated based on their information content and ontology graph position. As recommended in previous work (Guo *et al*, 2006), given a PPI involving two proteins, with A annotated by M terms and B annotated by N terms, we computed GO term similarity between all possible $M \times N$ term pairs and used the highest score as the SIM score between proteins A and B. Protein pairs involving a protein with no GO annotation do not receive an SIM score. We then compared the distribution of the similarity scores between our positive (Y2H) and random PPI sets. We created 100 random PPI networks by selecting 100 random worm genes from the entire genome as interactors for each of our 69 bait genes. SIM scores were calculated for all resulting random PPIs and these were compared with our Y2H data using a number of metrics: (i) the percentage of PPIs with a SIM score, (ii) the sum of all SIM scores, (iii) the average SIM score, which is the sum of SIM scores divided by the number of PPIs with an SIM score, and (iv) the overall average SIM score, which is the sum of SIM scores divided by the number of all PPIs (Supplementary Table 5).

## Previously published SH3 proteins involved in known PPIs

Experimental physical PPIs were retrieved from iRefWeb Release 3.4 (2 March 2011) (Turner *et al*, 2010). *C. elegans* SH3 protein Entrez Gene IDs were used in the search (Supplementary Table 6).

## GO gene function term enrichment and construction of the enrichment map

We used the DAVID v6.7 service to compute GO term enrichment for our yeast and worm interactomes (http://david.abcc.ncifcrf.gov/; Dennis *et al*, 2003; Huang da *et al*, 2009). The worm SH3 interactome contains 520 unique proteins, mapped to WormBase (WB) gene identifiers using WormMart (Schwarz *et al*, 2006). DAVID recognized 518 of these, which were used in subsequent DAVID functional analyses. The yeast SH3 interactome was previously published (Tonikian *et al*, 2009). DAVID recognized 370 yeast proteins which were used in DAVID functional analyses. The enrichment map in Figure 4 was built using the Enrichment Map Cytoscape Plugin (Merico *et al*, 2010). Functional modules were manually grouped and labeled using illustration software. The labels of each node (i.e., each functional category) are displayed in Supplementary Figure 13 (Figure 4).

## Protein ortholog identification and comparison of interactomes from different organisms

*H. sapiens* and *S. cerevisiae* orthologs of *C. elegans* proteins were retrieved from InParanoid (Berglund *et al*, 2008), OrthoMCL (Chen *et al*, 2006b), and Ensembl (Hubbard *et al*, 2009) and the union was taken (see Supplementary Table 19 for the full list of worm-to-yeast orthologs used in this work). Multiple orthologs were allowed.

## Protein interaction conservation

To probe the degree of SH3-mediated protein interaction conservation from worm to yeast and human, proteins from the worm SH3 domain interactome, restricted to interactions with SH3 domains as baits, were mapped to their yeast and human orthologs. Protein interactions in yeast were retrieved from the BioGRID database version 3.1.78 (Breitkreutz *et al*, 2008) and a previously published SH3 interactome (Tonikian *et al*, 2009). Protein interactions in human were retrieved from BioGRID version 3.1.78. When comparing worm with yeast, 98 proteins from the worm network had orthologs in yeast (Supplementary Table 19, only proteins involved in interactions in either worm or yeast are shown in Figure 5). To compute statistical significance of this overlap, the total number of possible interactions was estimated as the number of conserved baits (10) times the number of conserved preys (90). Since some baits can also act as preys, we subtracted $n_{bp} \times (n_{bp} - 1)/2$, where $n_{bp}$ is the number of conserved baits that also act as preys, to avoid counting the same interactions twice. In our data, $n_{bp}$ was equal to two, which gives a total of 899 possible interactions. Thus, finding two conserved interactions corresponds roughly to the overlap expected by chance ($P > 0.5$, Fisher's exact test, one-sided) (Figure 5; Supplementary Figure 3).

When comparing worm with human, 292 proteins from the worm SH3 domain interactome (restricted to SH3-mediated interactions) have orthologs in human (52 are baits and 251 are preys, $n_{bp} = 11$). In total, 536 worm PPIs from our interactome are present between these proteins and 265 human interactions could be mapped to worm by orthology, with an overlap of 31 conserved interactions between these two sets (Supplementary Figure 3). Although the fraction of conserved interactions is similar to the one between worm and yeast, the overlap is statistically significant ($P = 1.5 \times 10^{-7}$, Fisher's exact test). The different *P*-values come from the larger number of potential non-conserved interactions involving worm proteins with human orthologs.

## Binding motif definition

When predicting binding motifs on protein sequences to analyze rewiring scenarios or competitive interactions, we restricted the analysis to sequences that provide a clear match to the SH3 domain binding specificity. Binding motifs were defined as sequences with a PWM score larger than the threshold $T = 1000$, as defined previously. Consequently, only a subset of the best specificity matching motifs listed in Supplementary Table 3 are used as *bona fide* binding motifs in all subsequent analysis. Since a protein can have more than one stretch of amino acids matching a PWM, all ligand motifs with a PWM score larger than T were used, both in worm and in yeast. Therefore, in the

rewiring analysis, all motifs matching the binding specificity were retrieved and the most conserved motif in the other species was used to quantify motif conservation (Supplementary Table 9). When computing the total number of binding motifs per protein (Figure 6), motifs were clustered to ensure a minimal distance of at least 10 residues between two neighboring sites. As such, proteins with only one interacting SH3 domain in our network may have more than one potential binding motif (Figure 6). In total, 273 PPIs involving 24 SH3 domains and 164 proteins could be mapped at this level of detail.

## Endocytotic protein gene function prediction

Endocytotic protein gene function prediction was performed using a guilt-by-association strategy based on a modified k-core algorithm. Our expert-curated list of 109 worm endocytosis proteins (Supplementary Table 10) was used as the query. Modified k-cores were then defined as maximal sub-networks of proteins connected to at least one endocytosis protein and with *k* or more links between each other or to known endocytosis proteins. For example, a 1-core sub-network includes all worm proteins connected to at least one endocytosis protein. Other proteins in the worm SH3 domain interactome were assigned a score of zero, while worm proteins not present in the network were given the lowest score of $-1$. Ten-fold cross-validation was performed by dividing the list of known endocytosis genes into 10 sub-groups and alternatively using nine of them as the training set and one for testing. Negative example proteins (105 in total) used to compute ROC curves were randomly selected from the Worm proteome, excluding ones with known endocytosis function in any organism. ROC curves were computed for each cross-validation run by varying *k* and used to compute an average AUC of 0.69 ($P < 10^{-4}$). The k-core with $k = 3$ was used to make endocytosis predictions as it corresponded to a good ROC score, resulting in 86 novel proteins predicted to be involved in endocytosis (Supplementary Table 13).

## Data availability

Phage display data and other data from this paper are available for download from http://www.baderlab.org/Data/SH3Worm. Phospho-proteomics data are available from the PhosphoSitePlus database for the following cell line samples under the following accession numbers: LAN-6 End1, End2, and Lys, which can be accessed using the base URL: http://www.phosphosite.org/curatedInfoAction.do? record = 9391186 (CS6151), 9391899 (CS6152), 9391606 (CS6153, Akt substrate); and 9390610 (CS6121), 9391322 (CS6122), 9391006 (CS6123, pY); SMS-KCN End1, End2, Lys: 4148078 (CS5180), 4148082 (CS5181), 4148086 (CS5182, pY); 4318900 (CS5322), 4318902 (CS5323), 4318904 (CS5324, Akt substrate); TrkA expressing SK-N-BE(2) Lys, End1, End2: 15237598 (CS9939), 15237876 (CS9940), 15237832 (CS9941, Akt substrate); 18965770 (CS10550), 18965612 (CS10551), 18965530 (CS10552) (pY); Lys, End1: 15236716 (CS9203), 15236836 (CS9204, Akt substrate). The protein interactions from this publication have been submitted to the IMEx consortium (http://www.imexconsortium.org) through the IntAct database (Aranda *et al*, 2010) and assigned the identifier IM-18681.

## Supplementary information

## Acknowledgements

We thank Q Li, H Huang, J Gu, R Isserlin, H Fares, A Audhya, A Shi, S Pant, and D Carranza for experimental and computational support. This work was supported by the Canadian Institutes of Health Research (Grants MOP-84324 to GDB and MOP-93725 to SSS) and the US National Institutes of Health (Grants R01 GM067237 to BDG, R01 GM65462 to DGD and NHGRI R01 HG001715 awarded to MV and DEH). DG acknowledges the financial support of EMBO (EMBO-ALTF 241-2010) and SNSF (PBELA33-120936). The funders had no role in

study design, data collection and analysis, decision to publish, or preparation of the manuscript.

*Author contributions:* XX, DG, and GDB wrote the manuscript. XX and GDB led and coordinated the project. CB initiated the project. CB and GDB supervised the project. XX, DG, and GDB analyzed the data. DG performed the bioinformatics analysis. XX cloned yeast two-hybrid (Y2H) baits. XX and LL performed Y2H screens. J-FR helped with ORFeome Y2H screening. RT cloned phage display baits. AP and AL performed phage display assays. RT and SSS supervised the phage display assays. JC, ATC, and DB performed co-localization assays. DGD supervised co-localization assays. LS performed TBC-2 assay. AA obtained binding data for TBC-2 and AMPH-1. BDG designed and supervised the TBC-2 assay. AG and MLG performed the mass spectrometry assay. MLG analyzed the mass spectrometry data. MV and DEH provided human ORF clones, worm ORFeome Y2H library and supervised ORFeome Y2H screening. YZ, BC, SS, and TH performed sequencing of Y2H and phage display hits. XY, YAS, and KS-A helped with GFP cloning. JL helped with GO analysis.

## Conflict of interest

The authors declare that they have no conflict of interest.

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
