## [Review Process File · Molecular Systems Biology]

SH3 Interactome Conserves General Function Over Specific Form

Xiaofeng Xin, David Gfeller, Jackie Cheng, Raffi Tonikian, Lin Sun, Ailan Guo, Lianet Lopez, Alevtina Pavlenco, Adenrele Akintobi, Yingnan Zhang, Jean-François Rual, Bridget Currell, Somasekar Seshagiri, Tong Hao, Xiping Yang, Yun A. Shen, Kourosh Salehi-Ashtiani, Jingjing Li, Aaron T. Cheng, Dryden Bouamalay, Adrien Lugari, David E. Hill, Mark Grimes, David Drubin, Barth Grant, Marc Vidal, Charles Boone, Sachdev S. Sidhu, Gary D. Bader

Corresponding author: Gary Bader, University of Toronto

Review timeline:

Submission date:	10 October 2012
Editorial Decision:	02 November 2012
Revision received:	11 February 2013
Accepted:	20 February 2013

Editors: Andrew Hufton / Thomas Lemberger

Transaction Report:

1st Editorial Decision

02 November 2012

Thank you again for submitting your work to Molecular Systems Biology. We have now heard back from two of the three referees who agreed to evaluate your manuscript, and we have decided to render a decision now to avoid further delay. As you will see from the reports below, the referees find the topic of your study of potential interest. They raise, however, substantial concerns, which, I am afraid to write, must preclude its publication in its present form.

While the reviewers were cautiously positive about the goals of this work, the editor would like to emphasize that addressing their concerns appears to require additional experimental work (e.g. co-IP of AMPH-1/TBC-2 from live cells) and substantial new analyses. Reviewer #3 has important concerns with the evolutionary comparisons presented in this work, and feels that a more detailed analysis of binding specificity conservation is needed, as well as more cautious interpretations of the relative amounts of conservation between species pairs, given the low of number unambiguous orthologs in all comparisons. This reviewer also felt that a more thorough attempt should be made to place these findings in the context of other studies of evolution within interaction networks (the editor would also like to point a work by Habib et al, published very recently at our journal, doi:10.1038/msb.2012.50).

The first reviewer also suggested that you consider incorporating some of the supplementary figures and data into the main manuscript to improve readability. Our formatting requirement would allow you to include an additional 2-3 figures and/or tables in the main manuscript.

Please also deposit the new phospho-proteomic dataset in a public repository (e.g. PRIDE or PeptideAtlas), and include a confidential reviewer login with your revised manuscript.

If you feel you can satisfactorily deal with these points and those listed by the referees, you may wish to submit a revised version of your manuscript. Please attach a covering letter giving details of the way in which you have handled each of the points raised by the referees. A revised manuscript will be once again subject to review and you probably understand that we can give you no guarantee at this stage that the eventual outcome will be favorable.

Referee reports:

Reviewer #1 (Remarks to the Author):

The paper by Xin et al provides an extensive characterization of the repertoire of SH3 binding specificity and the SH3 interactome in *C. elegans*. These large-scale peptide phage display and yeast two-hybrid data sets are compared with each other as well as with related data sets from other organisms. These results suggest many novel SH3 interactions involved in endocytosis, which was supported in several cases by co-localization studies, and in one case (TBC-2) by more detailed follow-up studies. The most intriguing finding of this study was that although SH3 interactions are commonly involved in endocytosis both in the worm and in the yeast, the orthologous interactions are highly rewired.

This paper provides a significant advance in its field and deserves to be published in a prominent journal. However, some issues should be addressed before publication:

On page 6 the authors conclude that the Y2H data and the phage-derived binding specificities support each other. A more thorough analysis of this issue would be useful to justify this statement. If 63% of the Y2H hits were not among the top 500 predictions, an opposite argument could also be made. Since correlation with the phage display data was an inclusion criteria in filtering the Y2H data, the true overlap might be even less. While the lack of a possible binding site can obviously exclude binding (negative predictive value), it would be important to know what is the positive predictive value of a potential target site indicated by phage display to be favored among all the possible ones.

Many of the top-ranking SH3 ligand protein predictions provided in Supplementary Table 3 do not resemble closely the corresponding sequence logos provided in Figure 1, and instead often consist of long polyproline stretches. For example, the protein F59E12.9 was identified as a Y2H partner of *pix-1* SH3. This interaction was among the most highly ranked genome predictions (#4) because of the peptide region SSSPSPPPPPPPPPP. How can this be explained?

The AMPH-1 / TBC-2 interaction is verified using pull-down of in vitro translated TBC-2 with GST-AMPH1-SH3. This does not add much to the Y2H interaction already shown, and co-immunoprecipitation of these proteins from living cells would be much more convincing. While mislocalization of GFP-TBC-1 in *amph-1* deletion mutant worms is of interest, this may be true for many endocytosis-related proteins, and does not mean that a direct interaction between AMPH-1 and TBC-2 takes place in cells. However, this part of the story is not critical for the main message of the paper, and if it cannot be strengthened with reasonable efforts, these experiments could be excluded from this manuscript altogether.

If allowed by the space limitations of the journal, moving as many key data and figures as possible from the Supplementary Information into the manuscript itself would greatly facilitate reading of this paper.

Reviewer #3 (Remarks to the Author):

Xin and colleagues describe in this manuscript an effort to map the binding specificity and protein interactions of SH3 domains from *C. elegans*. They determined the binding specificity for 36 SH3 domains using phage display and identified 1070 protein interactions between 79 SH3 domains and

475 proteins using yeast-two-hybrid. These data were then used for an evolutionary comparative analysis between *C. elegans* and *S. cerevisiae* SH3 domains, using previously published data for yeast. From this analysis the authors conclude that the function of the SH3 mediated interactions are more likely conserved than the specific individual interactions. In addition, the authors predict novel protein functional associations, some of which were further validated experimentally. The topic and datasets generated for this work are very interesting. As far as I know this is the first effort to experimentally compare the binding specificity of a domain family beyond some previous work on the PDZ domain by some of these same authors (Tonikian et al. PLOS Bio 2008). However, some of the analysis is presented in very qualitative terms and can be substantially improved. Also, this study is very poorly contextualized with literally no mention of past work on the evolution of interaction networks. With significant improvements, this work would be a good fit for the audience of this journal.

Major concerns:

- Given the wealth of data obtain for this project some of the analysis is somewhat lacking. In particular, when the authors compare the binding specificity of the SH3 domains between the two species (*C. elegans* and *S. cerevisiae*) they merely compare the fraction of the SH3 domains that match class I/II/atypical in both species. Even in their previous study of PDZ domains these same authors were much more particular in comparing the specificity of 6 orthologous domains (Figure 6 of Tonikian et al. PLOS Bio 2008). The authors should improve significantly this comparative analysis. At the very least they should attempt to compare the specificity of different classes of homology across these 2 species (ex. 1-to-1 orthologous domains, many-to-many orthologous domains, cases of duplication in one of the species, etc). What is the fraction of orthologous domains that have diverged considerably in their binding specificity and does that vary with gene or domain duplication ? Are there changes in protein properties that could explain or be correlated with changes in specificity for orthologous domains (e.g. changes in domain composition, lower abundance when compared with proteins with domains having conserved specificity, gene duplication events, etc) ? Do paralogous worm SH3 domains have very similar specificity as it was the case for *S. cerevisiae* (e.g. *Boi1* and *Boi2*, *Myo3* and *Myo5*, etc).

- Related to the point above, the analysis of the conservation/divergence of function for the SH3 interactions in both species could also be improved. It appears clear from their data that there is some functional conservation with additional species specific functions but there is very little analysis and discussion of the origin of the species specific functional roles. Are these due to : 1) interactions between conserved SH3 domains with species specific proteins; 2) species specific SH3 domains; 3) new binding sites within conserved proteins ?

- Although the manuscript is focused on the evolution of SH3 mediated protein-protein interactions there is literally no mention of previous related work anywhere in the manuscript. There is one citation to the interolog paper (Matthews et al. 2001) but only in relation to the benchmarking the Y2H interactions. As the authors are well aware there is a large body of work now on the evolution of different types of protein-protein interactions (complexes versus linear-motifs) and previous evidence supporting a fast divergence of linear-motif mediated interactions. In fact, it is not surprising that these SH3 mediated interactions are so divergent but well in line with what past studies have discovered. It is even more perplexing when some of the same authors (Charles Boone) have authored papers specifically about the evolution of SH3 interactions (Sun et al. PLOS Comp Bio 2012) that were not cited. Considering that there is now over a decade of literature on the evolution of cellular interaction networks the authors should really place this work in the context of previous studies.

Minor comments:

- In their previous study of SH3 mediated interactions in yeast (Tonikian et al. PLOS Bio 2009) the authors used a machine learning approach to integrate the PWMs information with the protein-protein interactions from experiments. Was there a specific reason why this was not done again in this study (lack of true-positive set, other) ?
- How did the authors define a threshold value for the PWMs in order to predict potential binding sites ? They mentioned a threshold T but no benchmark test or numerical rationale for how it might have been selected.
- When comparing the conservation of the SH3 interactions between worm, yeast and human I am not convinced that there is a substantial difference in conservation between yeast/worm and

human/worm. In fact the fraction of conserved interactions is similar in both cases ($2/37=0.054$ in yeast and $31/536=0.057$ for human). Given the small number of available cases for the yeast to worm analysis (37) I don't think the authors can make a strong case for the statement that SH3 mediated interactions are more conserved between human and worm than between yeast and worm.

- In order to test the predictions for endocytosis related functional annotations the authors have, among other things, isolated lysosomes and endosomes that were metabolically labelled and subjected these to mass-spec analysis. It is very peculiar that a phospho-peptide enrichment step was used when the objective was to identify relative abundance differences of full proteins. The authors never justify the use of this phospho-peptide enrichment step.
- The results section, in particular the data analysis methods, often have description of results and even discussion. These could be moved to the main results and discussion or simply removed when redundant.
- Please avoid the term "primitive" in "more primitive yeast" (page 15). Use for example less complex or unicellular.

1st Revision - authors' response

11 February 2013

Thank you for organizing the careful and useful review of our manuscript. We are delighted to see that the two reviewers recognize the importance, novelty and general interest of our work. The reviewers make excellent points, which we address below and in a substantially revised manuscript. First, we have expanded the functional analysis of rewired interactions, which further supports our model of core endocytosis proteins used to recruit more specialized endocytosis protein in an organism specific manner. Second, we provide a much more detailed analysis of our data, including showing the complementarity of Y2H and phage display results at the domain level. Third, we have positioned the manuscript in the context of other studies of protein interaction network evolution by expanding the introduction and discussion sections and including the missing references. You can find below a detailed reply to all reviewers comments.

Following the comments of reviewer 1, we have incorporated two Figures and several text sections previously in supplementary information in the main manuscript.

The phospho-proteomics data are already publicly available in the PhosphoSitePlus database (list of URLs in the supplementary information section "4. Data availability" and the protein interactions have been submitted to the IntAct database and made available in the standard PSI-MI data format on the supplementary data website.

We hope the improved manuscript will be suitable for publication. Thank you again for your help in reviewing our manuscript.

Reviewer #1 (Remarks to the Author):

The paper by Xin et al provides an extensive characterization of the repertoire of SH3 binding specificity and the SH3 interactome in C. elegans. These large-scale peptide phage display and yeast two-hybrid data sets are compared with each other as well as with related data sets from other organisms. These results suggest many novel SH3 interactions involved in endocytosis, which was supported in several cases by co-localization studies, and in one case (TBC-2) by more detailed follow-up studies. The most intriguing finding of this study was that although SH3 interactions are commonly involved in endocytosis both in the worm and in the yeast, the orthologous interactions are highly rewired.

This paper provides a significant advance in its field and deserves to be published in a prominent journal.

We thank the reviewer for this positive and encouraging evaluation of our work.

However, some issues should be addressed before publication:

On page 6 the authors conclude that the Y2H data and the phage-derived binding specificities support each other. A more thorough analysis of this issue would be useful to justify this statement. If 63% of the Y2H hits were not among the top 500 predictions, an opposite argument could also be made.

We thank the reviewer for this important remark. To investigate this issue, we now provide a detailed comparison analysis for each domain separately (Supplementary Figure S2). We observe that for the majority of domains the enrichment remains true with an excellent P-value ($P < 0.001$). For 10 out of the 36 domains, the enrichment was not as significant ($P\text{-value} > 0.001$), suggesting that Y2H data do not fully agree with phage predictions. These 10 domains are characterized by a much lower number of interactions in our network (4.8 versus 15.3). This is interesting because it shows that well-connected SH3 domains in our network are not merely sticky proteins in our Y2H data, further supporting our results. It could be that some of these SH3 domains with fewer interactions have evolved different binding properties, have another role in vivo than simply mediating protein interactions, and therefore are not fully characterized by the in vitro interaction profile that we map with phage display, or depend on weaker interactions that are not well measured by phage display (which is limited to detection of strong/optimal interactions).

From a more general point of view, Y2H was stringently carried out using well-established methods and is expected to identify many true positive interactions (as we show by comparison with multiple independent data sources). Further, Y2H and phage display are orthogonal in their respective strengths and weaknesses, thus are expected to identify different aspects of protein interactions. For instance, phage display may not capture all aspects of SH3 binding, such as weak binding, co-operative binding, non-canonical binding modes or those that involve larger binding sites than the peptides used here (12mers). Also, we don't have phage display data for all SH3 domains mainly due to the difficulty of purifying correctly folded GST-fusion SH3 domains. So just because there is no overlap with a phage display predicted interaction, does not mean the Y2H data is false. We use the statistically significant agreement between the data sets to strengthen both data sets – if two independent data sets significantly agree, they are more likely to be correct.

>Since correlation with the phage display data was an inclusion criteria in filtering the Y2H data, the true overlap might be >even less.

We apologize for not being clear on this point in the previous version of the manuscript. In all enrichment analysis, we have been very careful not to include any circularity. For the overlap mentioned here, we have not considered any interaction that was included based on the PWM score in the comparison. The manuscript has been updated to better describe this detail. Also, the number of interactions added only because of the presence of a sequence motif matching the SH3 binding specificity (i.e. interactions with only one evidence in the Y2H but with a high PWM score) is low (<7%) compared to the total number of interactions and thus doesn't play a major role in Y2H filtering.

While the lack of a possible binding site can obviously exclude binding (negative predictive value), it would be important to know what is the positive predictive value of a potential target site indicated by phage display to be favored among all the possible ones.

We first would like to point out that the lack of binding site does not necessarily exclude physiologically relevant binding (phage display has some limitations, as discussed above). Regarding the predictive value of phage data, we agree that simply finding sequence hits can result in many false-positives, especially since poly-proline regions typically provide good hits, while not all of them are likely to bind to all SH3 domains. This is the main reason why we used the phage predictions mainly to include Y2H interactions with only one line of evidence (e.g. one colony instead of two). However, we note that several studies (e.g. Tonikian et al. PLoS Biology 2009, Tong et al. Science 2002, Landgraf et al. PLoS Biology 2004, Carducci et al. Biotechnol Adv. 2012) have shown that target sites predicted based on the domain specificity are often correct and that a higher accuracy is achieved in protein interaction prediction by combining different evidences, which is the main rationale for our strategy to combine Y2H and phage data.

Many of the top-ranking SH3 ligand protein predictions provided in Supplementary Table 3 do not resemble closely the corresponding sequence logos provided in Figure 1, and instead often consist of long polyproline stretches. For example, the protein F59E12.9 was identified as a Y2H partner of pix-1 SH3. This interaction was among the most highly ranked genome predictions (#4) because of the peptide region SSSPSPPPPPPPPPP. How can this be explained?

Long poly-proline stretches are often ranked very high because they match the specificity in many positions. Several of these are known to interact with SH3 domains (e.g. searching PubMed for “(poly-proline OR polyproline) AND SH3” finds many papers e.g. Urbanek et al. Curr Biol. 2012 which studies a good example in Las17/WASP). In the case of pix-1, the specificity profile contains three positions that are highly specific for proline and four others where proline is the most or second most often observed amino acids. Finally the serine is also favored at the position where the last serine is found in the F59E12.9 sequence. Therefore this polyproline region corresponds to a good motif match. In this case we agree that the arginine characteristic of class I and known to play an important role from structural studies is absent, however none of the RxxPxxP motifs in F59E12.9 contains the proline two residues upstream of the arginine, which is also known to be important, based on the phage profile. As we decided to adopt an unbiased strategy using only the experimental phage data to define the binding specificity, the resulting rank for SSSPSPPPPPPPPPP is quite high and based on our experience, we expect this to be a reasonable approach.

The AMPH-1 / TBC-2 interaction is verified using pull-down of in vitro translated TBC-2 with GST-AMPH1-SH3. This does not add much to the Y2H interaction already shown, and co-immunoprecipitation of these proteins from living cells would be much more convincing. While mislocalization of GFP-TBC-1 in amph-1 deletion mutant worms is of interest, this may be true for many endocytosis-related proteins, and does not mean that a direct interaction between AMPH-1 and TBC-2 takes place in cells. However, this part of the story is not critical for the main message of the paper, and if it cannot be strengthened with reasonable efforts, these experiments could be excluded from this manuscript altogether.

The AMPH-1 / TBC-2 interaction is not part of the Y2H network itself, but was predicted from the phage data alone (we apologize that this was not clear). So the in vitro binding that we showed provided experimental support for an otherwise hypothetical binding interaction. Second, in a previous paper by members of our team (Pant et al. Nature Cell Biology 2009) the effects of loss of amph-1 on six different endocytosis proteins was tested. Only the one that was shown to bind directly to AMPH-1 (i.e. RME-1) was lost from the membrane. The other five endocytosis proteins (SDPN-1, clathrin, dynamin, RAB-5, RAB-10) all remained membrane bound in amph-1 deletion mutants. Thus loss of amph-1 does not result in a large-scale redistribution of endosome proteins to the cytoplasm, as suggested by the reviewer. We considered performing the requested co-IP, but in our previous experience, these rarely work in worms because worms are difficult to lyse and the harsh lysis conditions tend to disrupt weaker protein interactions. As a result only a few in vivo co-IPs in worm have been successfully published. We included this section to show the utility of the data set (in this case the phage display derived binding specificity of the AMPH-1 SH3 domain) in predicting interactions in worm at the binding site level and to take advantage of the data to make a discovery about AMPH-1 function. These points are now clarified in the text.

If allowed by the space limitations of the journal, moving as many key data and figures as possible from the Supplementary Information into the manuscript itself would greatly facilitate reading of this paper.

We moved more experimental methods and data analysis description and figures to the main text: the Y2H/phage overlap (formerly figure S1) and the worm protein expression control (formerly figure S5 – now integrated into the worm TBC-2 localization experiment figure) were moved and a new figure comparing ortholog and paralog SH3 domain sequence specificity was added as requested by reviewer 3.

Reviewer #3 (Remarks to the Author):

Xin and colleagues describe in this manuscript an effort to map the binding specificity and protein interactions of SH3 domains from C. elegans. They determined the binding specificity for 36 SH3

domains using phage display and identified 1070 protein interactions between 79 SH3 domains and 475 proteins using yeast-two-hybrid. These data were then used for an evolutionary comparative analysis between C. elegans and S. cerevisiae SH3 domains, using previously published data for yeast. From this analysis the authors conclude that the function of the SH3 mediated interactions are more likely conserved than the specific individual interactions. In addition, the authors predict novel protein functional associations, some of which were further validated experimentally. The topic and datasets generated for this work are very interesting. As far as I know this is the first effort to experimentally compare the binding specificity of a domain family beyond some previous work on the PDZ domain by some of these same authors (Tonikian et al. PLOS Bio 2008). However, some of the analysis is presented in very qualitative terms and can be substantially improved. Also, this study is very poorly contextualized with literally no mention of past work on the evolution of interaction networks. With significant improvements, this work would be a good fit for the audience of this journal.

We thank the reviewer for acknowledging the interest and the novelty of our work and agree that several analyses and a broader literature review were missing in the previous version. We have substantially revised our manuscript to address these issues.

Major concerns:

- Given the wealth of data obtain for this project some of the analysis is somewhat lacking. In particular, when the authors compare the binding specificity of the SH3 domains between the two species (C. elegans and S. cerevisiae) they merely compare the fraction of the SH3 domains that match class I/II/atypical in both species. Even in their previous study of PDZ domains these same authors were much more particular in comparing the specificity of 6 orthologous domains (Figure 6 of Tonikian et al. PLOS Bio 2008). The authors should improve significantly this comparative analysis.

At the very least they should attempt to compare the specificity of different classes of homology across these 2 species (ex. 1-to-1 orthologous domains, many-to-many orthologous domains, cases of duplication in one of the species, etc). What is the fraction of orthologous domains that have diverged considerably in their binding specificity and does that vary with gene or domain duplication? Are there changes in protein properties that could explain or be correlated with changes in specificity for orthologous domains (e.g. changes in domain composition, lower abundance when compared with proteins with domains having conserved specificity, gene duplication events, etc)? Do paralogous worm SH3 domains have very similar specificity as it was the case for S. cerevisiae (e.g. Boi1 and Boi2, Myo3 and Myo5, etc).

We thank the reviewer for this very useful point. We have now included a detailed comparison of SH3 domain specificity between ortholog and paralog domains in yeast and worm (new Figure 2). The conclusion that the repertoire of binding specificity is conserved is maintained, but we now also observe that SH3 domains on orthologous proteins have often diverged at the sequence level and, for several of them, also at the specificity level. We see a similar pattern for the worm paralogs that we have phage data for, in contrast to the highly conserved specificity of yeast paralogs (e.g. Boi1 and Boi2, Myo3 and Myo5). There is one case of one-to-many (Hum-1 <-> Myo3+Myo5), most likely because of a gene duplication in yeast and one case with domain duplication in yeast (Sdpn-1 <-> Bzz1#1 + Bzz1#2). The low level of SH3 domain specificity conservation between orthologs is interesting. In particular, it confirms our observation of frequent rewiring that is due both to changes/loss in domain specificity and motif changes/loss. In general, specificity conservation/divergence seems to be correlated with the sequence similarity of the SH3 domains, with domains that are more sequence similar being more likely to have conserved specificity (see new supplementary Figure S1 which compares sequence similarity to PWM similarity). This additional more detailed result agrees with the overall observed rewiring at the protein interaction level. We have added a new section describing these results. We are currently collecting phage display data for human SH3 domains and this additional data should enable us to study this system in more detail in future work.

- Related to the point above, the analysis of the conservation/divergence of function for the SH3 interactions in both species could also be improved. It appears clear from their data that there is some functional conservation with additional species specific functions but there is very little analysis and discussion of the origin of the species specific functional roles. Are these due to: 1)

interactions between conserved SH3 domains with species specific proteins; 2) species specific SH3 domains; 3) new binding sites within conserved proteins?

This is another useful point – thank you. We have now included an additional analysis of the functional conservation of endocytosis at the level of single SH3 domains and of their interactions. The worm SH3 interactome contains 42 endocytosis proteins. Only ten have conserved endocytosis function in yeast (denoted as ‘functionally conserved’). Thus, the majority of worm endocytosis proteins in our network are novel compared to yeast. Eleven of these novel endocytosis proteins directly interact with at least one of the ten functionally conserved worm endocytosis proteins. Focusing specifically on SH3-containing proteins, the worm SH3 interactome contains 15 endocytosis SH3-containing proteins. Six have yeast orthologs, and four (HUM-1, ITSN-1, AMPH-1, SDPN-1) are functionally conserved in yeast. Interestingly, these four proteins mediate 8 interactions with worm endocytosis proteins that are not functionally conserved in yeast (point (1)), and only 2 interactions with functionally conserved worm endocytosis proteins. Similarly, four of the eleven (=15 - 4) worm endocytosis SH3 proteins not functionally conserved in yeast interact with at least one functionally conserved worm endocytosis protein (point (2)). This suggests that novel worm endocytosis proteins often acquire their endocytosis function through interactions with functionally conserved core endocytosis proteins. More generally, of the 34 worm SH3 interactome PPIs involving two endocytosis proteins, only two involve proteins with yeast orthologs that are both involved in endocytosis: ITSN-1 (Ede1p in yeast) and SDPN-1 (Bzz1p in yeast) interacting with DYN-1 (Vps1p in yeast). The majority of interactions include a pair of worm proteins with one (15 PPIs) or two (17 PPIs) yeast orthologs not known to act in endocytosis, or not conserved. These observations support our model where some core endocytosis proteins are conserved between different organisms and are used to recruit other proteins in an organism specific manner, conferring new endocytosis functions to proteins that do not have this function in the other organism.

We are aware that our analysis mainly deals with endocytosis while there are many proteins with other biological functions in our network. However the other functions generally have a lower number of proteins in our data, which makes statistical analysis less confident. Therefore we decided to focus on endocytosis for which more abundant and reliable data were available.

- Although the manuscript is focused on the evolution of SH3 mediated protein-protein interactions there is literally no mention of previous related work anywhere in the manuscript. There is one citation to the interolog paper (Matthews et al. 2001) but only in relation to the benchmarking the Y2H interactions. As the authors are well aware there is a large body of work now on the evolution of different types of protein-protein interactions (complexes versus linear-motifs) and previous evidence supporting a fast divergence of linear-motif mediated interactions. In fact, it is not surprising that these SH3 mediated interactions are so divergent but well in line with what past studies have discovered. It is even more perplexing when some of the same authors (Charles Boone) have authored papers specifically about the evolution of SH3 interactions (Sun et al. PLOS Comp Bio 2012) that were not cited. Considering that there is now over a decade of literature on the evolution of cellular interaction networks the authors should really place this work in the context of previous studies.

We apologize for this oversight. We have now better contextualized the work with a thorough literature review, addressed with added context in the introduction and a substantially expanded section in the discussion. Please let us know if we missed important work.

Minor comments:

- In their previous study of SH3 mediated interactions in yeast (Tonikian et al. PLOS Bio 2009) the authors used a machine learning approach to integrate the PWMs information with the protein-protein interactions from experiments. Was there a specific reason why this was not done again in this study (lack of true-positive set, other)?

Yes, the reason we did not follow a machine learning approach is the lack of a representative gold-standard of positive (and negative) interactions in worm. Additionally, we only had two types of data (Y2H and phage display) where Tonikian et al. also had peptide array data, and our data coverage was not as complete as for the yeast work (i.e. we only have phage data for 1/3 of worm SH3 domains). Therefore using a machine learning approach is difficult and won't guarantee better results.

- How did the authors define a threshold value for the PWMs in order to predict potential binding sites? They mentioned a threshold T but no benchmark test or numerical rationale for how it might have been selected.

Thanks for this comment. We have now included more detail about how the threshold was carefully chosen. The threshold T was established following two different strategies. First we observed that the median value of the PWM scores for Y2H interactions with more than one evidence is equal to 972. Second, by manually inspecting the predicted motifs, it appeared quite clearly that most motifs with scores larger than 1000 matched the binding specificity well, while most motifs with lower scores were poor matches. Therefore we decided to choose a threshold of $T=1000$. We also note that only 67 out of a total of 1070 interactions were added because they showed a good PWM score with $Y2H_count=1$. Thus only a small fraction of the network depends on this threshold value. Moreover, varying this threshold by two orders of magnitude (i.e. from 1000 to 10 or to 100,000) did not alter the number of conserved interactions both in yeast and in human.

- When comparing the conservation of the SH3 interactions between worm, yeast and human I am not convinced that there is a substantial difference in conservation between yeast/worm and human/worm. In fact the fraction of conserved interactions is similar in both cases ($2/37=0.054$ in yeast and $31/536=0.057$ for human). Given the small number of available cases for the yeast to worm analysis (37) I don't think the authors can make a strong case for the statement that SH3 mediated interactions are more conserved between human and worm than between yeast and worm.

This is an excellent point that we have now clarified in the text. The statistical analysis was done assuming a random model where interactions are distributed randomly, but where the total number of interactions is that observed experimentally. As the number of possible non-conserved interactions between worm proteins with human orthologs is much larger compared to the worm-yeast comparison, the same fraction of conserved interactions give rise to very different P-values in the two comparisons. We have clarified this point in the manuscript. Although the P-value analysis suggests some level of conservation not observed between worm and yeast, we agree that this is still a speculative statement that cannot be definitely proven based on our data. We've modified the conclusion accordingly. Nevertheless, we note that this is not a completely unexpected conclusion, since higher level of conservation between worm and human is also observed for many biological properties (e.g. many more worm/human orthologs exist in the SH3 interactome).

- In order to test the predictions for endocytosis related functional annotations the authors have, among other things, isolated lysosomes and endosomes that were metabolically labelled and subjected these to mass-spec analysis. It is very peculiar that a phospho-peptide enrichment step was used when the objective was to identify relative abundance differences of full proteins. The authors never justify the use of this phospho-peptide enrichment step.

This data set was already collected and available to us and provided additional experimental evidence for our endocytosis gene function prediction (in particular based on very careful fractionation of endosomes), thus we included it. We have clarified this in the text.

- The results section, in particular the data analysis methods, often have description of results and even discussion. These could be moved to the main results and discussion or simply removed when redundant.

We have tried to move all of the results and discussion from methods to the relevant results and discussion section in the main text. Please let us know if you notice additional specific ones we missed.

- Please avoid the term "primitive" in "more primitive yeast" (page 15). Use for example less complex or unicellular.

Good point. Text has been updated to use 'unicellular'.